# Retrograde transport of TrkB-containing autophagosomes via the adaptor AP-2 mediates neuronal complexity and prevents neurodegeneration

Natalia L. Kononenko[1,2,3,*], Gala A. Claßen[1,*], Marijn Kuijpers[1], Dmytro Puchkov[1], Tanja Maritzen[1], Aleksandra Tempes[4], Anna R. Malik[4], Agnieszka Skalecka[4], Sujoy Bera[3], Jacek Jaworski[4] & Volker Haucke[1,2,5]

Autophagosomes primarily mediate turnover of cytoplasmic proteins or organelles to provide nutrients and eliminate damaged proteins. In neurons, autophagosomes form in distal axons and are trafficked retrogradely to fuse with lysosomes in the soma. Although defective neuronal autophagy is associated with neurodegeneration, the function of neuronal autophagosomes remains incompletely understood. We show that in neurons, autophagosomes promote neuronal complexity and prevent neurodegeneration *in vivo* via retrograde transport of brain-derived neurotrophic factor (BDNF)-activated TrkB receptors. p150[Glued]/dynactin-dependent transport of TrkB-containing autophagosomes requires their association with the endocytic adaptor AP-2, an essential protein complex previously thought to function exclusively in clathrin-mediated endocytosis. These data highlight a novel non-canonical function of AP-2 in retrograde transport of BDNF/TrkB-containing autophagosomes in neurons and reveal a causative link between autophagy and BDNF/TrkB signalling.

[1] Leibniz-Institut für Molekulare Pharmakologie, 13125 Berlin, Germany. [2] NeuroCure Cluster of Excellence, Charité Universitätsmedizin Berlin, 10117 Berlin, Germany. [3] CECAD Research Center, University of Cologne, 50931 Cologne, Germany. [4] Laboratory of Molecular and Cellular Neurobiology, International Institute of Molecular and Cell Biology, 02-109 Warsaw, Poland. [5] Freie Universität Berlin, Faculty of Biology, Chemistry and Pharmacy, 14195 Berlin, Germany. * These authors contributed equally to this work. Correspondence and requests for materials should be addressed to N.L.K. (email: n.kononenko@uni-koeln.de) or to V.H. (email: haucke@fmp-berlin.de).

Autophagy is an evolutionary conserved process that serves to provide nutrients during starvation and to eliminate defective proteins and organelles[1,2] such as mitochondria and the endoplasmic reticulum via lysosomal degradation[3]. During autophagy portions of the cytoplasm are sequestered within double- or multimembraned vesicles termed autophagosomes. These undergo subsequent maturation steps, in particular fusion with late endosomes, to become late-stage autophagosomes also called amphisomes[4] before being delivered to lysosomes by dynein-mediated retrograde transport[5,6]. Autophagosome formation requires an E3-like complex comprising ATG5 that catalyses lipid conjugation of microtubule-associated protein 1 light chain 3 (LC3) (ref. 2).

In addition to the cytoprotective function of autophagy under conditions of starvation[7], recent data support additional roles of autophagy, for example, in maintenance of stemness[8] or FGF signalling to mediate bone growth during development[9]. In the brain, autophagosomes form locally in distal axons and are trafficked retrogradely[10] to eventually fuse with lysosomes enriched in the neuronal soma. Accumulation of autophagosomes is a hallmark of neurodegenerative disorders including Alzheimer's and Huntington's disease, or amyotrophic lateral sclerosis[11–14], while knockout (KO) of key autophagy proteins in mice causes neurodegeneration[15,16]. In spite of these findings the physiological function of neuronal autophagosomes and their role in promoting neuronal survival and counteracting neurodegeneration remains incompletely understood.

A crucial pathway that promotes neuronal survival, protects from neurodegeneration and promotes neuronal complexity[17] is the brain-derived neurotrophic factor (BDNF) signalling pathway. In cortical and hippocampal neurons BDNF initiates signalling by binding to its receptor TrkB in distal neurites[17]. Activated BDNF/TrkB complexes are internalized predominantly via macropinocytosis mediated by EHD4/pincher into so-called 'signalling endosomes' that are refractory to lysosomal degradation to ensure persistent signalling[18]. Consistent with this model, BDNF/TrkB have been shown to require retrograde axonal transport to promote neuronal branching and survival and to counteract neurodegeneration[19,20]. Recent data suggest that TrkB-signalling endosomes may contain late endosomal markers such as Rab7 (ref. 21) and are trafficked in part by Snapin, a subunit of the BLOC-1 complex. KO mice lacking Snapin suffer from impaired neurosecretion, but do not show major defects in brain architecture or neuronal complexity[22] associated with defective BDNF/TrkB signalling, suggesting that other factors must exist that promote retrograde traffic of TrkB-signalling endosomes. However, neither the identity of these factors nor the cell biological nature of TrkB-signalling endosomes is known.

Here we demonstrate that TrkB-signalling endosomes are late-stage autophagosomes that undergo retrograde transport to the neuronal soma via their association with the adaptor AP-2, an essential[23] protein complex hitherto thought to function exclusively in clathrin-mediated endocytosis[24,25] and in the reformation of synaptic vesicles in the brain[26]. AP-2 is a heterotetramer comprised of α, β, μ, and σ subunits. We show that neuronal AP-2 mediates retrograde transport of TrkB-containing autophagosomes via association of AP-2α with LC3 and of AP-2β with the p150$^{Glued}$ subunit of the dynein cofactor dynactin to promote neuronal complexity and counteract neurodegeneration *in vivo*. Our data, thus, identify a novel function of autophagy in BDNF/TrkB signalling during brain development mediated by a non-canonical role of the endocytic adaptor AP-2.

## Results

**Retrograde co-trafficking of AP-2 and LC3 in neurons**. The localization and function of neuronal AP-2 so far has been investigated largely by analysis of its steady-state distribution in fixed neurons and brain tissue or by genetic and biochemical experiments[23–26]. To obtain insights into AP-2 function in living neurons we monitored the dynamics of fluorescent protein-tagged AP-2μ (tagged at an internal site that retains full functionality[27,28]) by live imaging of primary cortico-hippocampal neurons in culture. In addition to the plasma membrane, AP-2μ-mRFP localized to puncta in axons and dendrites (Fig. 1a,b). In axons, AP-2μ-mRFP-positive puncta underwent rapid bidirectional movement (at $0.4$–$0.5\,\mu m\,s^{-1}$, Fig. 1c, Supplementary Movie 1) suggestive of microtubule-based transport with a preference for retrograde transport to the neuronal soma (Fig. 1d). In contrast, no retrograde transport of AP-2μ-mRFP was seen in dendrites (Supplementary Fig. 1a,b, Supplementary Movie 2). Given the reported association of AP-2 with autophagic proteins[29,30] and the prominent microtubule-based movement of autophagosomes in neurites[5,10], we hypothesized that mobile AP-2-containing structures may correspond to LC3-positive autophagosomes. Live cell image analysis revealed a close colocalization (Supplementary Fig. 1c,d) and co-trafficking (Fig. 1e,f, Supplementary Movie 3) of eGFP-LC3b with AP-2μ-mRFP in primary neurons. Pearson's correlation analysis showed that the degree of colocalization between AP-2μ-mRFP and eGFP-LC3 (Rp = 0.65 ± 0.05) (Fig. 1g) was comparable to that of eGFP-LC3 with mCherry-ATG12 (Rp = 0.67 ± 0.02), a *bona fide* component of the machinery for autophagosome formation (Supplementary Fig. 1e,f). In agreement with previous reports, we found that >80% of stationary AP-2 puncta in the axon were confined to synapses (Supplementary Fig. 1g–i). Partial co-localization of endogenous AP-2 with LC3 in neuronal processes was further confirmed by dual-colour time-gated stimulated emission depletion microscopy (STED) analysis of neurons treated with the vATPase inhibitor folimycin, a lysosomotropic agent, which prevents autophagosome, and, thus, LC3b degradation via lysosomal proteolysis, and immunostained for AP-2α and LC3b (Fig. 1h,i, Supplementary Fig. 1j,k). In clathrin-mediated endocytosis, AP-2 functions to recruit clathrin and cargo proteins to endocytic sites at the plasma membrane. Surprisingly, we failed to detect a specific enrichment of clathrin on AP-2-positive autophagosomes (Supplementary Fig. 1l,m). These data suggest that retrograde co-trafficking of AP-2 on autophagosomes may reflect a novel non-canonical function of AP-2 in autophagosome transport in primary neurons that likely is independent of its established role in clathrin-mediated endocytosis as further discussed below.

**AP-2 forms a complex with LC3 and dynactin/p150$^{Glued}$**. The co-trafficking of AP-2 with LC3 on autophagosomes raises the question how neuronal AP-2 is recruited to these carriers. While the β, μ and σ subunits of AP-2 complex are made from a single gene in mammals, the α subunit is encoded by two isogenes termed $\alpha_A$ and $\alpha_C$ that undergo alternative splicing in the brain. Recent data suggest that AP-2 can associate with LC3 via a putative LIR motif within the appendage domain of the AP-2$\alpha_A$ subunit[29]. To probe whether AP-2 via its α-appendage domain directly binds to LC3 we carried out binding assays using purified proteins (Fig. 2a, Supplementary Fig. 2a,c). Purified recombinant LC3 was found to bind to the GST-tagged appendage domains of both AP-2α and AP-2$\alpha_C$ with a preference for AP-2$\alpha_A$ over AP-2$\alpha_C$ (Fig. 2a). In line with this, endogenous AP-2$\alpha_{A/C}$ co-purified with GST-LC3b (Supplementary Fig. 2c) in affinity

chromatography experiments from brain lysates (Fig. 2b), while, conversely, GST-AP-2α$_A$ and, less well, GST-AP-2α$_C$ associated with native LC3b (Supplementary Fig. 2b). Consistent with the preferential retrograde transport of AP-2-containing LC3-positive autophagosomes, we found endogenous AP-2 to co-immunoprecipitate with the p150$^{Glued}$ subunit of dynactin, a cofactor for the retrograde microtubule-based motor dynein, but not with the anterograde trafficking motor Kif5A from detergent-extracted rat brain lysates (Fig. 2c). Moreover, p150$^{Glued}$ expressed in HEK293T cells was able to capture the purified appendage domain of AP-2β in binding assays (Fig. 2d). Finally, affinity purification of endogenous AP-2 using GST-fused Stonin 2 as a bait resulted in the co-purification of both LC3b and p150$^{Glued}$ (Fig. 2e, see also Supplementary Fig. 2d), suggesting that all three proteins act as part of a complex. Consistent with

these biochemical data we observed the close colocalization of endogenous LC3b with p150$^{Glued}$ and AP-2 in neurons treated with folimycin (Supplementary Fig. 2e–h). Collectively, these findings indicate that AP-2 associates with LC3 and p150$^{Glued}$ in a protein complex that may mediate retrograde transport of autophagosomes.

**AP-2 regulates retrograde autophagosome transport in neurons.** To test this hypothesis we probed whether AP-2 is functionally important for retrograde autophagosome transport by conditionally ablating expression of the essential AP-2μ subunit[23] in neurons[26]. Tamoxifen addition to DIV0 cultured hippocampal neurons isolated from newborn mice carrying floxed alleles of AP-2μ and expressing a tamoxifen-inducible Cre recombinase

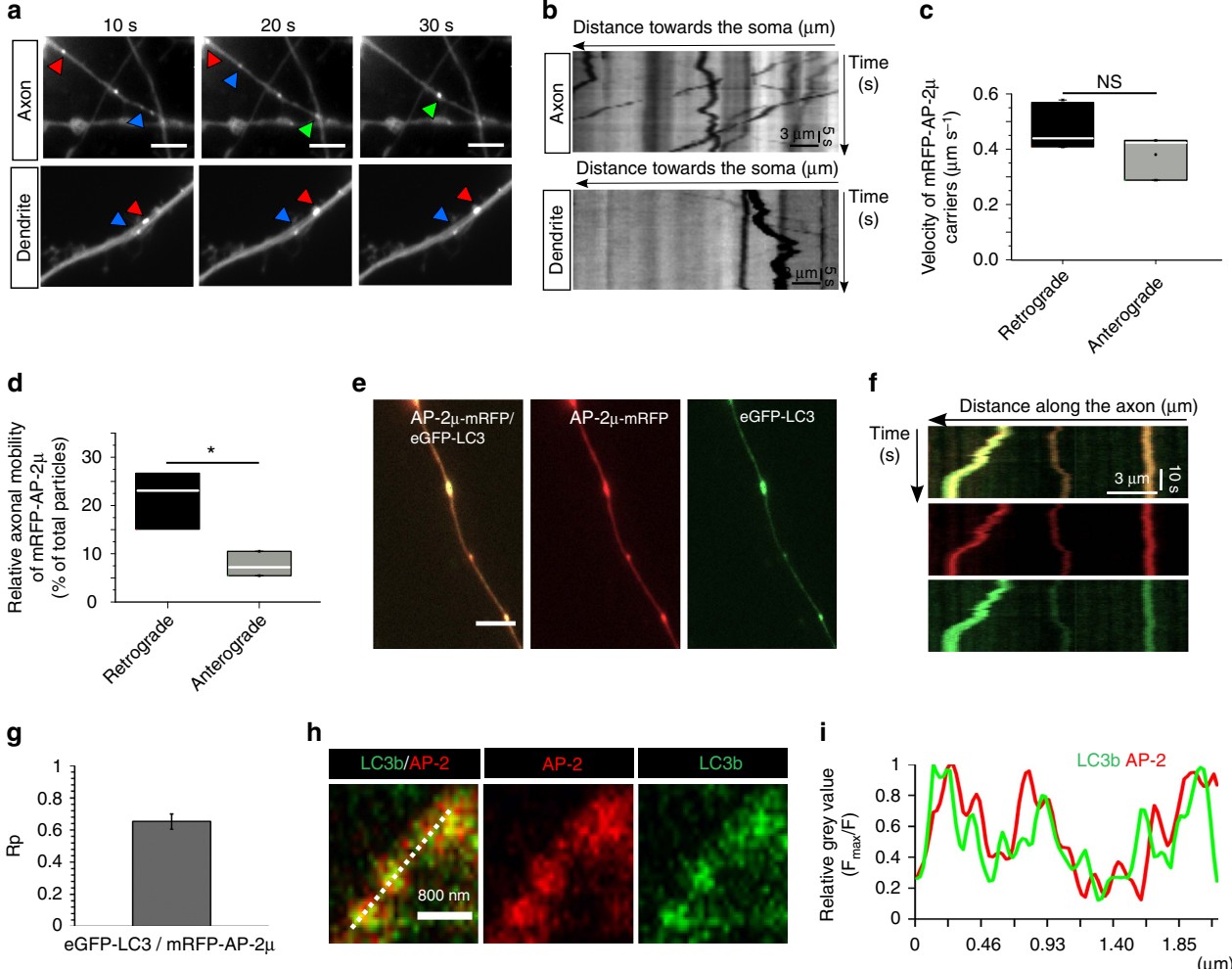

**Figure 1 | Retrograde co-trafficking of adaptor protein complex 2 (AP-2) and LC3 in cultured cortico-hippocampal neurons.** (a–d) Dynamics of AP-2μ-mRFP transport in neurons. (a) Representative time series and corresponding kymographs (b) of AP-2μ-mRFP-positive puncta in axons and dendrites in control neurons. Coloured arrows in a indicate retrograde carriers. See also overview in Supplementary Fig. 1a. Scale bars, 5 μm. (c) Mean retrograde and anterograde velocities of AP-2μ-mRFP-positive carriers in axons (retrograde: 0.47 ± 0.04 μm s$^{-1}$, anterograde: 0.38 ± 0.04 μm s$^{-1}$, $P = 0.246$, $n = 3$ independent experiments). NS, non-significant. (d) AP-2μ-mRFP carriers preferentially undergo retrograde movement (21.6 ± 3%) in comparison to anterograde movement (7.7 ± 1.3%, *$P = 0.02$, $n = 3$ independent experiments) in axons. (e–g) AP-2μ-mRFP associates with autophagosomes in axons. (e,f) Representative time series and corresponding kymographs showing the colocalization (e) and cotransport (f) of AP-2μ-mRFP with eGFP-LC3b-labelled autophagosomes. (g) Bar diagram representing the colocalization of AP-2μ-mRFP with eGFP-LC3b in neurons based on Pearson's coefficient (Rp) (Rp: 0.65 ± 0.05). Rp was calculated for 26 regions of interest (ROI, 5.1 × 4.3 μm) per condition from $n = 3$ independent experiments. Scale bar in e, 5 μm. (h,i) Time-gated stimulated emission depletion (STED) microcopy analysis of endogenous AP-2 and LC3 localization in neurons acutely treated with folimycin to inhibit lysosomal proteolysis of autophagosomes (20 nM, 4 h). Representative images (h) and corresponding line scans (i) of neuronal processes immunostained for endogenous AP-2α and LC3b. See also overview in Supplementary Fig. 1j. Scale bar in h, 800 nm. Data in c,d are illustrated as box plots as described in Methods. Data in g and all data reported in the text are mean ± s.e.m.

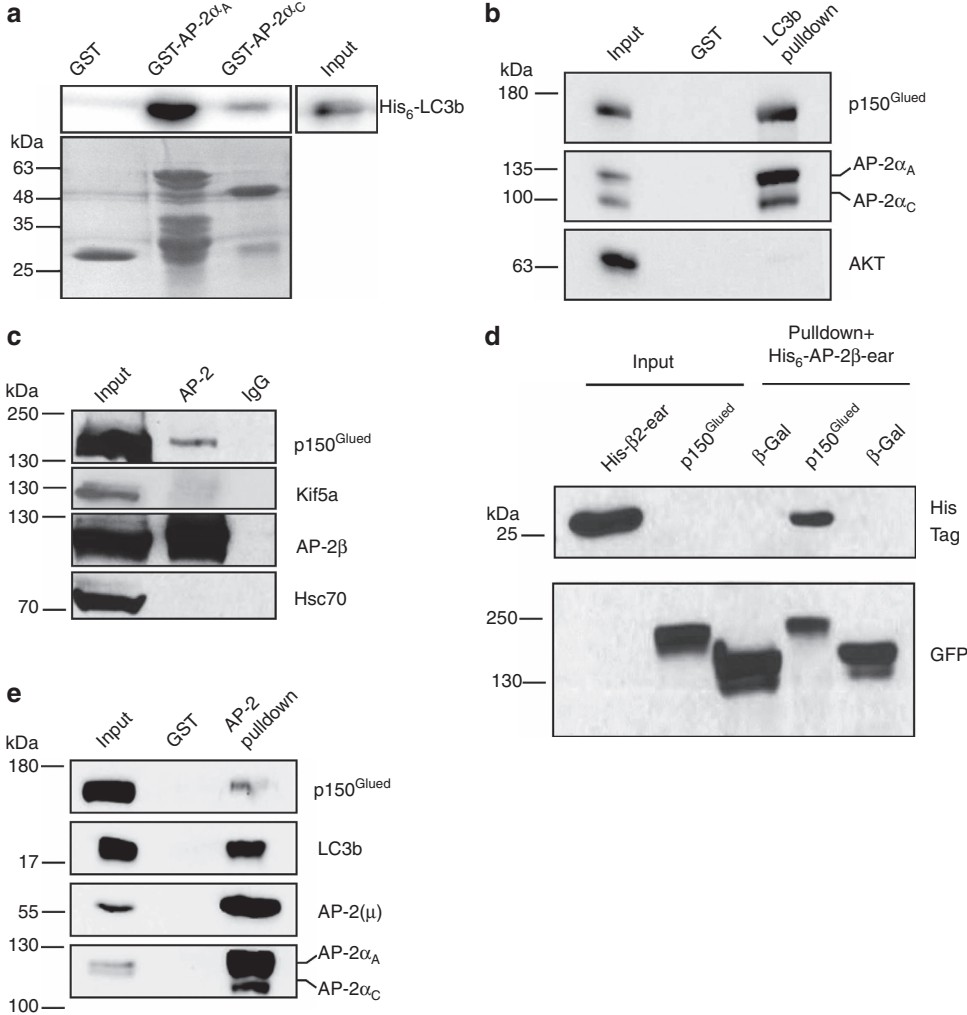

**Figure 2 | Direct binding of AP-2α to LC3b and association with p150^Glued/dynactin. (a)** Upper panel: purified recombinant LC3b-His$_6$ detected by immunoblotting directly binds to GST-AP-2α$_A$ with preference (about threefold) over GST-AP-2α$_C$. Input, 2% of the total recombinant LC3b added to the assay. Lower panel: ponceau-stained membrane. Example from $n = 3$ independent experiments. **(b)** Complex formation of neuronal AP-2 with LC3b and the p150^Glued subunit of dynactin. Affinity purifications of mouse brain lysates using GST-LC3b as bait co-purified AP-2α$_A$ and, to a lesser extent AP-2α$_C$, as well as p150^Glued, but not the negative control protein Akt. Input, 1.5% of lysate added to the assay. Representative example from $n = 4$ independent experiments. **(c)** Co-immunoprecipitation of endogenous AP-2 with p150^Glued, but not with Kif5A from rat brain lysate using antibodies against AP-2β. Hsc70, used as a negative control, was not co-precipitated. Input, 10% of lysate added to the assay. Representative example from $n = 3$ independent experiments. **(d)** AP-2β-ear interacts with p150^Glued. AviGFP-tagged p150^Glued or β-galactosidase were expressed together with BirA in HEK293T cells, affinity-purified using M-280 streptavidin Dynabeads, incubated with recombinant His$_6$-AP-2β-ear and analysed by immunoblotting. Input, 10% of lysate added to the assay. Representative example from $n = 3$ independent experiments. **(e)** AP-2 interacts with LC3b and p150^Glued. Affinity purifications using a GST-tagged N-terminal fragment of Stonin 2 as bait co-purified endogenous AP-2 (detected via its μ and α subunits), LC3b and p150^Glued. Input, 1.4% of lysate added to the assay. Representative example from $n = 3$ independent experiments.

(AP-2μ^lox/lox:CAG-iCre) resulted in strongly reduced expression of neuronal AP-2 (monitored by its α subunit) (Supplementary Fig. 3a,b). The levels of other endocytic or presynaptic proteins analysed in brain lysates from conditional neuronal-confined AP-2 KO neurons (AP-2μ^lox/lox:Tubα1-Cre mice; described in more detail below) were largely unaffected except for a small, yet statistically insignificant reduction of synaptotagmin 1 (Supplementary Fig. 3c,d), an established AP-2 binding partner[31].

We then monitored the transport of autophagosomes in wild-type (WT) and AP-2μ KO neurons expressing mRFP-eGFP-LC3 by live imaging. In WT neurons mRFP-labelled autophagosomes displayed bidirectional movements with an average retrograde velocity of about 0.4–0.5 μm s$^{-1}$ (Fig. 3a–c), similar to the values obtained for AP-2μ-mRFP with which it colocalizes (compare Fig. 1) and consistent with earlier data[10,32]. AP-2μ deletion

greatly reduced retrograde autophagosome velocity and the mobile fraction of retrograde LC3b-positive carriers (Fig. 3a–c, Supplementary Fig. 3e), while the fraction of stationary LC3b puncta was increased (Supplementary Fig. 3f). In agreement with the function of dynein motors in slow anterograde movement, lack of neuronal AP-2 caused a mild, yet statistically insignificant reduction in anterograde autophagosome transport (Supplementary Fig. 3e,g)[33]. Transport of mitochondria proceeded unperturbed in absence of neuronal AP-2 (Supplementary Fig. 3h–k). These data suggest that AP-2 regulates retrograde transport of autophagosomes from neurites to the cell soma, where most lysosomes are located[5,10]. Consistent with this hypothesis and with our live imaging results analysis of AP-2μ KO neurons by thin-section electron microcopy revealed an accumulation of dense vesicular and concentric multilamellar organelles

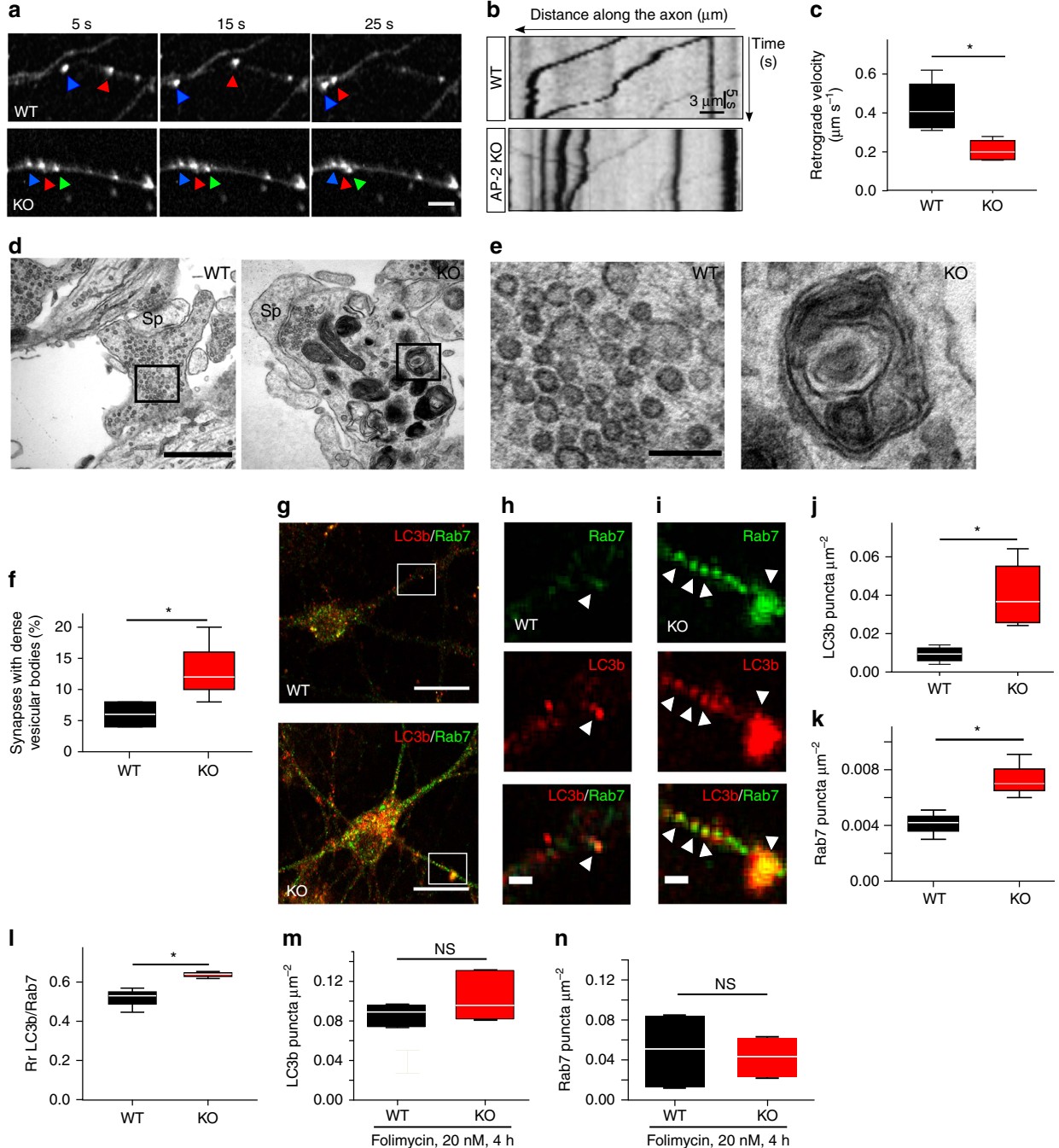

**Figure 3 | AP-2 regulates autophagosome transport in neurons.** (**a**) Time-lapse images of mRFP-LC3-positive puncta (arrows) in WT and AP-2μ KO neurons. Scale bar, 5 μm. (**b**) Kymographs of mRFP-LC3 carriers generated from **a**. (**c**) Average retrograde velocity of mRFP-LC3 carriers in WT and AP-2μ KO neurons. Loss of AP-2μ significantly decreased the LC3 velocity compared to WT controls (WT: $0.44 \pm 0.07\,\mu m\,s^{-1}$, KO: $0.21 \pm 0.03\,\mu m\,s^{-1}$, *$P = 0.019$, $n = 4$ independent experiments, 49–67 neurites per condition). (**d**–**e**) Electron micrographs of synapses from cultured WT and AP-2μ KO neurons. AP-2 KO neurons accumulate dense vesicular bodies with the majority representing concentric multilamellar structures (black boxes in **d** represent magnified areas in **e**). Scale bars, (**d**) 500 nm, (**e**) 100 nm. Sp, spine. See also Supplementary Fig. 3l–o. (**f**) Percentage of WT and KO synapses containing dense vesicular bodies (WT: $6.00\% \pm 1.54\%$, AP-2 KO: $13.00\% \pm 2.52\%$, *$P = 0.045$, $n = 4$, 100 synapses per condition). (**g**–**i**) Representative confocal images of cultured WT and AP-2 KO neurons immunostained for LC3b and Rab7 (white boxes in **g** represent the magnified areas in **h**,**i**). Scale bars, (**g**) 15 μm, (**h**,**i**) 2 μm. (**j**,**k**) Accumulation of LC3b-containing structures (LC3b puncta $\mu m^{-2}$ are depicted, WT: $0.009 \pm 0.002$, AP-2 KO: $0.040 \pm 0.009$, *$P = 0.016$, $n = 4$, in total 39 AP-2 KO and 33 WT neurons) (**j**) and Rab7-containing structures (Rab7 puncta $\mu m^{-2}$ are depicted, WT: $0.004 \pm 0.000$, AP-2 KO: $0.007 \pm 0.000$, *$P = 0.042$, $n = 3$, in total 29 AP-2 KO and 23 WT neurons) (**k**) in AP-2μ-KO neurons. (**l**) Enhanced colocalization of LC3b with Rab7 on neuronal autophagosomes in absence of AP-2μ based on Pearson's coefficient (Rp) (WT: $0.52 \pm 0.04$, AP-2 KO: $0.64 \pm 0.01$, *$P = 0.032$). Rp was calculated for 64–84 regions of interest (ROI) per condition from three independent experiments ($n = 3$). (**m**,**n**) Bar diagrams indicating similar numbers of LC3b- (WT: $0.09 \pm 0.01$, AP-2 KO: $0.1 \pm 0.02$) (**m**) and Rab7-positive puncta $\mu m^{-2}$ (WT: $0.05 \pm 0.02\%$, AP-2 KO: $0.04 \pm 0.01$) (**n**) in WT and AP-2μ KO neurons treated with folimycin.. Shown is the number of puncta per $\mu m^2$ ($n = 3$ independent experiments, 26 neurons per condition). Data in **c**,**f**,**j**–**n** are illustrated as box plots as described in Methods. Data reported in the text are mean ± s.e.m. NS, non-significant.

resembling late-stage autophagosomes (also termed amphisomes) post-fusion with late endosomes (Fig. 3d–f, Supplementary Fig. 3o and below). As reported earlier[26], AP-2μ KO neurons displayed a reduced number of synaptic vesicles (to about 60% of those seen in WT), in agreement with its canonical function in synaptic vesicle reformation. The 'spheroid-like' accumulation of late-stage autophagosomes in neurites and in the soma of AP-2μ KO compared to WT control neurons was confirmed by immunostaining with antibodies against endogenous LC3b and late endosomal Rab7 (Fig. 3g–l for quantifications), indicating that the autophagosomal structures observed by light and electron microscopy have undergone fusion with late endosomes. In contrast, no significant alterations in the number or localization of early endosomes marked by Rab5 (Supplementary Fig. 4a,b) or LAMP1-positive late endosomes/lysosomes (Supplementary Fig. 3l–n, Supplementary Fig. 4c,d) were observed. To probe whether loss of AP-2 may alter LC3b synthesis and/or degradation[34,35], we treated WT and KO neurons with the lysosomotropic agent folimycin. Folimycin application caused the accumulation of LC3b-positive puncta in WT neurons to reach levels similar to those seen in neurons lacking AP-2μ (Fig. 3m, Supplementary Fig. 4e). Thus, LC3b accumulation in absence of AP-2 does not appear to result from increased LC3b synthesis. A similar increase in folimycin-treated AP-2 KO neurons was observed for Rab7-positive puncta (Fig. 3n). These results argue that the accumulation of late-stage autophagosomes in AP-2 KO neurons is caused by decreased degradation of LC3b/Rab7-positive autophagosomes due to their defective retrograde transport.

**AP-2 regulates autophagy independent of endocytosis**. As decreased degradation of LC3b/Rab7-positive autophagosomes in the absence of AP-2μ might result from their defective delivery to lysosomes, we next probed the turnover of autophagosomes in the absence of AP-2μ using mRFP-eGFP-LC3 as a reporter (Fig. 4a). Serum deprivation caused an elevation of the mRFP/eGFP fluorescent ratio in WT neurons indicative of increased starvation-induced autophagic flux. In contrast, no significant changes in the mRFP/eGFP ratio were observed in AP-2μ KO neurons (Fig. 4b, Supplementary Fig. 4f). Moreover, AP-2 loss resulted in the accumulation of the autophagic adaptor and degradative substrate p62/SQSTM1 (Fig. 4c,d) that persisted upon inhibition of protein synthesis in the presence of cylcoheximide (Supplementary Fig. 4g), suggesting that it is the result of defective autophagosome turnover. Consistent with this, mTORC1 signalling measured by phospho-S6 kinase 1 and phospho-Raptor levels was not significantly changed in AP-2 KO brains, although we detected a slight increase in the total amount of S6 kinase (Supplementary Fig. 4h,i). These data suggest that autophagosome accumulation in absence of AP-2 is not a consequence of reduced mTORC1 activity, a key repressor of autophagosome formation.

Our results described thus far indicate that loss of AP-2 causes defective retrograde transport and accumulation of autophagosomes in neurons, likely via its association with p150[Glued] and with LC3. A prediction from this hypothesis is that AP-2 binding to LC3 is required for autophagosome transport and turnover. To test this directly we capitalized on a mutant variant of AP-2α_A that due to mutational inactivation of its LIR motif fails to associate with LC3 (ref. 29). Overexpression of LC3 binding-deficient mutant, but not WT AP-2α_A in control neurons phenocopied AP-2μ loss with respect to defective mRFP-eGFP-LC3 conversion and retrograde autophagosome transport (Fig. 4e,f). By contrast, mutant AP-2α_A was perfectly capable of restoring defective clathrin-mediated endocytosis of

transferrin in HeLa cells depleted of endogenous AP-2α_A and AP-2α_C (Fig. 4g,h). Collectively, these data reveal a novel function for AP-2 in retrograde axonal transport of LC3/Rab7-containing autophagosomes in neurons that appears to be independent of its established role in endocytosis.

**Autophagosome transport by AP-2 promotes neuronal complexity**. What is the function of LC3/Rab7-containing autophagosomes trafficked retrogradely via complex formation of AP-2 with LC3? Previous data show that Rab7-positive organelles act as retrograde shuttling devices for active neurotrophins including BDNF and its main receptor TrkB (refs 36,37) to mediate nuclear signalling[38] and neuronal arborization[39]. To test this, we monitored the retrograde dynamics of TrkB-mRFP by live imaging. TrkB-mRFP was present on both large, as well as smaller diffraction-limited mobile puncta within the soma and in neurites of BDNF-stimulated neurons, where it colocalized with eGFP-LC3-positive autophagosomes (Fig. 5a,b; Supplementary Movie 4). In the absence of BDNF the mobility of TrkB-mRFP puncta was greatly reduced in WT (Fig. 5c,d), but not in AP-2 KO neurons (Supplementary Fig. 5a,b), suggesting that BDNF may induce the targeting of signalling-active TrkB to autophagosomes for AP-2/LC3/p150[Glued]-mediated transport. Consistent with this hypothesis we observed that TrkB-mRFP, as well as the active phosphorylated form of TrkB (pTrkB) and the TrkB signalling component growth factor receptor-bound protein 2 (Grb2) accumulated in LC3b-containing autophagosomes in the absence of AP-2μ (Fig. 5e–g, Supplementary Fig. 5c–f). The levels of full-length TrkB and its truncated isoform T1 were elevated in brains of conditional AP-2μ KO mice (Supplementary Fig. 5g,h), which persisted upon inhibition of protein synthesis by cylcoheximide (Supplementary Fig. 5i,j), suggesting that it is the consequence of impaired turnover of stalled TrkB-containing autophagosomes. The levels of the p75NGF receptor, which can also be activated by BDNF, but is not coupled to the BDNF/TrkB signalling pathway were unaltered (Supplementary Fig. 5k–m). Importantly, AP-2μ loss significantly reduced the mobility of TrkB-mRFP puncta (Fig. 5h–j), in line with their retrograde traffic via autophagosomes. Endocytosis of active TrkB proceeded in the absence of AP-2μ (Supplementary Fig. 5n–p), consistent with TrkB being internalized predominantly via EHD4/pincher-mediated macropinocytosis[18] and with an endocytosis-independent role of AP-2 in retrograde transport of TrkB-containing autophagosomes. Collectively, these data indicate that AP-2 mediates retrograde transport of LC3-positive autophagosomes containing active TrkB.

Retrograde trafficking of TrkB from axons to the neuronal soma is required for neurotrophin signalling to promote neuronal arborization and complexity[39]. Hence, defective retrograde transport of TrkB-containing autophagosomes in the absence of AP-2 would be expected to result in reduced neuronal complexity and impaired dendritic arborization. We directly tested this in hippocampal neurons from AP-2μ[lox/lox]:CAG-iCre mice, in which AP-2μ expression can be acutely abrogated by tamoxifen treatment. As predicted neuronal complexity was severely reduced in tamoxifen-treated AP-2 KO neurons (treated from DIV0) compared to mock-treated WT control neurons (Fig. 5k,l), a phenotype rescued by re-expression of AP-2μ at DIV8 (Fig. 5m, Supplementary Fig. 6a–c). Moreover, expression of LC3 binding-deficient AP-2α_A (Fig. 5n, Supplementary Fig. 6d,e) or partial depletion of the p150[Glued] subunit of dynactin (Supplementary Fig. 6f–i) in WT control neurons both phenocopied decreased neuronal complexity observed in neurons lacking AP-2μ, suggesting that retrograde axonal transport of TrkB-containing

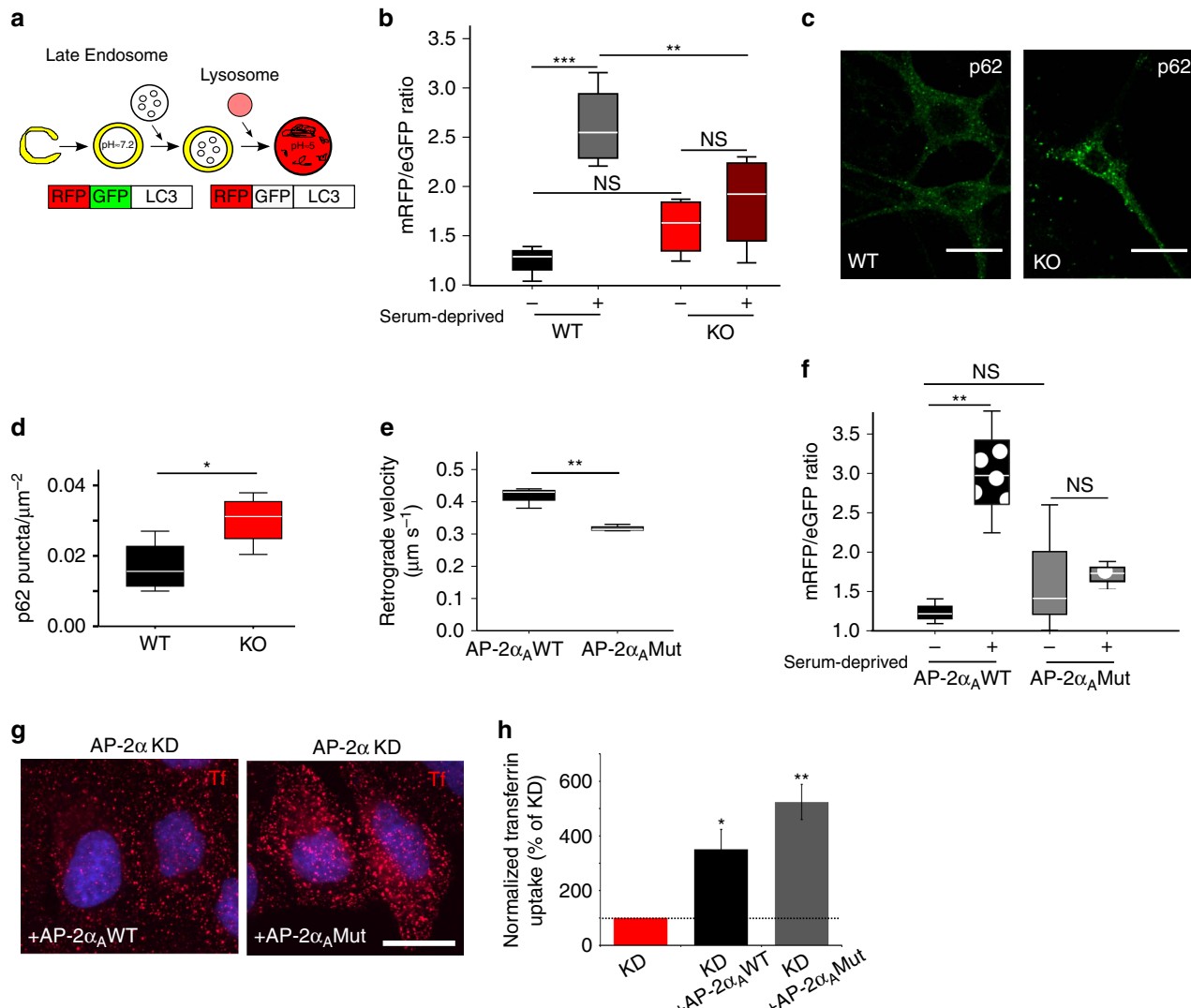

**Figure 4 | AP-2 regulates autophagosome turnover independent of its role in endocytosis. (a)** Tandem mRFP-eGFP-tagged LC3 as a reporter of autolysosome formation. **(b)** Mean mRFP/eGFP intensity ratio in control or serum-deprived WT or AP-2μ KO neurons ($n = 6$ independent experiments, with 37–54 neurons per condition). No significant difference between WT and AP-2μ KO neurons was observed at steady state ($P = 0.398$). Serum deprivation failed to trigger the formation of autolysosomes in AP-2μ neurons (control WT: 1.247 ± 0.092, serum-deprived WT: 2.847 ± 0.213, \*\*\*$P < 0.001$, control KO: 1.528 ± 0.130, serum-deprived KO: 1.774 ± 0.169, $P = 0.640$; serum-deprived WT versus serum-deprived KO \*\*$P = 0.002$). **(c)** Representative confocal images of WT and AP-2μ KO neurons immunostained for p62. Scale bars, 20 μm. **(d)** Increased number of p62-positive puncta per μm² in AP-2μ KO (0.030 ± 0.003) compared to WT neurons (0.017 ± 0.003). \*$P = 0.046$, $n = 4$, 33–39 neurons per condition. See also Supplementary Fig. 4g. **(e)** Average retrograde velocity of mRFP-LC3 carriers in control neurons expressing AP-2$\alpha_A$ WT or LC3 binding-deficient AP-2$\alpha_A$ Mut and co-expressing mRFP-eGFP-LC3. Loss of LC3-AP-2α binding significantly decreased LC3 transport compared to AP-2$\alpha_A$ WT expressing controls (AP-2$\alpha_A$WT: 0.42 ± 0.00 μm s$^{-1}$, AP-2$\alpha_A$Mut: 0.32 ± 0.01 μm s$^{-1}$, \*\*$P = 0.007$, $n = 3$ independent experiments, ≥45–47 neurites per condition). **(f)** Mean mRFP/eGFP intensity ratio in control or serum-deprived neurons expressing AP-2$\alpha_A$WT or AP-2$\alpha_A$Mut ($n = 4$ independent experiments, ≥25 neurons per condition). No significant difference between neurons expressing AP-2$\alpha_A$WT or AP-2$\alpha_A$Mut was observed at steady state ($P = 0.661$). Serum deprivation fails to trigger autolysosome formation in neurons expressing AP-2$\alpha_A$Mut (control AP-2$\alpha_A$Mut: 1.544 ± 0.361, serum-deprived AP-2$\alpha_A$Mut: 1.775 ± 0.088, $P = 0.886$), but not in neurons expressing AP-2$\alpha_A$WT (control AP-2$\alpha_A$WT: 1.173 ± 0.091, serum-deprived AP-2$\alpha_A$WT: 2.660 ± 0.240, \*\*$P = 0.003$). **(g,h)** LC3-binding defective AP-2$\alpha_A$ (AP-2αA Mut) restores clathrin-mediated endocytosis of transferrin in HeLa cells depleted of endogenous AP-2α (KD) (\*$P = 0.041$, \*\*$P = 0.007$, $n = 4$, 156, 124, 148 cells per condition, respectively). Mean grey values of transferrin uptake in KD + AP-2$\alpha_A$WT and KD + AP-2$\alpha_A$Mut conditions were normalized to KD condition set to 100%. Scale bar, 20 μm. Tf, transferrin. Data in **b,d,e,f** are illustrated as box plots as described in Methods. Data in **h** and all data reported in the text are mean ± s.e.m. NS, non-significant.

autophagosomes mediated by an AP-2/LC3/p150$^{Glued}$ complex is required for proper neuronal arborization. To test whether the loss of neuronal branching in DIV14 neurons results from a failure in neurite development or whether it is a consequence of dendrite pruning, we induced AP-2μ loss in fully developed neurons, by applying tamoxifen in cultured eGFP-expressing neurons at DIV8 (Supplementary Fig. 6j). Conditional loss of

AP-2μ induced at DIV8 significantly impaired the neuronal branching complexity of mature neurons (Supplementary Fig. 6k,l). These data suggest that loss of AP-2-mediated retrograde autophagosome transport induces post-developmental neurite pruning in cultured neurons.

A prediction from these data is that defective autophagosome maturation and, thus, impaired shuttling of TrkB signals should

result in reduced neuronal complexity, akin to loss of AP-2 or p150[Glued] function. To test the prediction that reduced neuronal complexity is causally linked to defective TrkB-containing autophagosome transport, we conditionally deleted ATG5, an E3 ligase mediating early autophagosome expansion and maturation[40], via tamoxifen-induced recombination in neurons from ATG5[lox/lox]:CAG-iCre mice (ATG5 KO). Conditional loss of ATG5 in hippocampal neurons in culture resulted in defective neuronal arborization (Fig. 5o,p, Supplementary Fig. 6m). We conclude that AP-2/LC3-mediated retrograde transport of

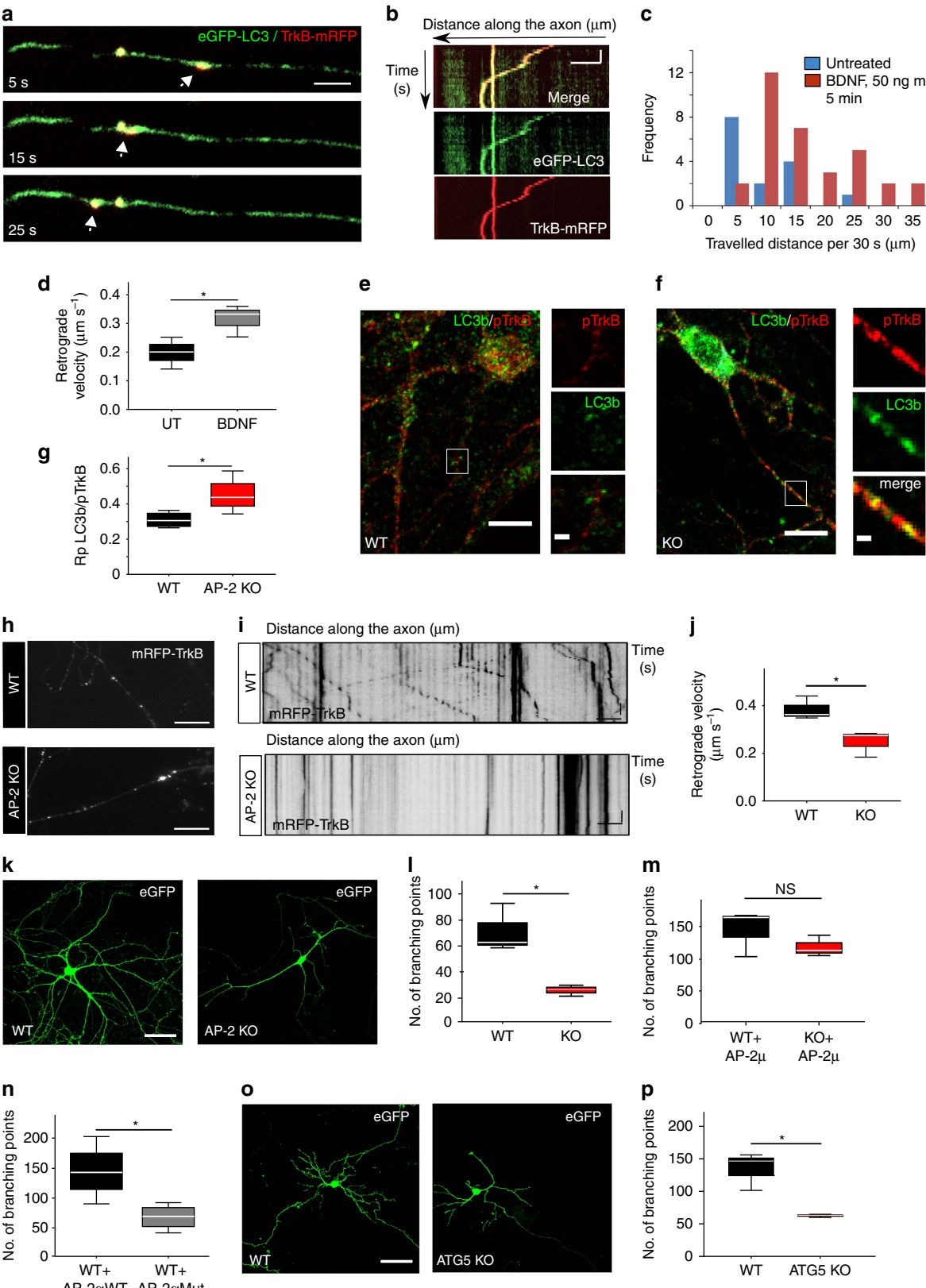

TrkB-containing autophagosomes promotes neuronal complexity *in vitro*.

Given the prominent defects in transport of TrkB-containing autophagosomes and neuronal arborization observed in hippocampal neurons in culture following acute inactivation of AP-2μ, we wanted to explore the functional importance of AP-2 for TrkB-mediated neuronal arborization in the brain *in vivo*. To specifically ablate AP-2μ expression in CNS neurons we crossed floxed AP-2μ mice with a strain expressing Cre under the neuron-specific tubulin α1 promoter[41] starting from embryonic day 13.5 (AP-2μ[lox/lox]:Tubα1-Cre mice). Conditional AP-2μ KO mice were born well below Mendelian ratios (KO: 16% instead of 25% as expected, $P < 0.0011$, see Supplementary Fig. 7a) and lagged behind in postnatal development including cessation of weight gain at about 2 weeks and postnatal lethality between postnatal day (p) 21 and 26 (Fig. 6a,b, Supplementary Fig. 7b). To analyse the role of AP-2 in dendritic arborization *in vivo* we visualized the branching complexity of neurons in the cortex of p20 control and AP-2 KO mice by Golgi silver impregnation. Neuron-specific AP-2μ KO mice displayed dramatic defects in dendritic architecture compared to WT littermates (Fig. 6c–f), as evident from Sholl analysis of stellate cells in cortical layers III and II, respectively (Fig. 6g,h). These data confirm our observations from cultured AP-2μ KO neurons and suggest that neuronal AP-2 is required for neuronal arborization *in vivo*.

**Neuronal AP-2 prevents neurodegeneration *in vivo*.** Histopathological analysis of Nissl-stained brain sections from AP-2μ KO mice at p20 further revealed a marked degeneration of the thalamus (Fig. 6i), including micro-vacuolations within the thalamic neuropil indicative of spongiform neurodegeneration (Supplementary Fig. 7c). Fluoro-Jade staining, a probe that detects dying neurons, further confirmed the neurodegeneration in brains of conditional AP-2μ KO mice at p20 (Supplementary Fig. 7d). Loss of neurons in absence of AP-2 was mediated by apoptosis, as evidenced by the marked elevation of active caspase-3 in lysates from cultured AP-2 KO neurons (Supplementary Fig. 7e,f). Histological analysis of the temporal progression of neurodegeneration in conditional AP-2μ KO revealed that although the brain morphology of AP-2μ-deficient mice appeared normal at p4 (Supplementary Fig. 7g,h,m), already at p7 first signs of neuronal loss were detectable in thalamic nuclei (Supplementary Fig. 7i,j,n). This was followed by the dramatic appearance of spongiform neurodegeneration in the cortex at p14 (Supplementary Fig. 7k,l,o). These data suggest that AP-2μ is required to prevent neuronal loss and neurodegeneration.

As thalamic afferents profoundly affect postnatal development of the somatosensory cortex[42], we analysed the morphology of barrel compartments in WT and AP-2μ-deficient mice. Barrel compartments were absent in Nissl-stained brain sections of AP-2μ KO mice (Supplementary Fig. 7q), a phenotype, which was confirmed by immunostaining for potassium-chloride transporter member 2 (KCC2) (Fig. 6j). These morphological and gross anatomical alterations were accompanied by an accumulation of p62/SQSTM1 (Fig. 6k–m) and of LC3b/Rab7-containing puncta, likely corresponding to stalled late-stage autophagosomes (Supplementary Fig. 7r–u).

Collectively, these results confirm our observations from cultured AP-2μ KO neurons and suggest that neuronal AP-2 is required for neuronal arborization and to prevent neurodegeneration *in vivo*. In agreement with this model impaired trafficking of autophagosomes[6,43–45] or defective BDNF/TrkB signalling are associated with neurodegeneration[19,20].

**Impaired BDNF-TrkB signalling in AP-2-deficient neurons.** The data described thus far suggest a model according to which AP-2 promotes retrograde transport of LC3/Rab7-positive autophagosomes containing active TrkB complexes to convey distal BDNF signals to the soma[46] and thereby promote neuronal complexity (Fig. 7a). A major target of TrkB signalling is BDNF itself, the expression of which has recently been found to be under the control of a TrkB-mediated positive feedback loop[47–49]. Thus, if AP-2 loss indeed impairs TrkB signalling one would expect the expression of BDNF to be reduced in KO mice lacking neuronal AP-2. In agreement with this hypothesis, we found pro-BDNF and BDNF levels to be reduced by about 50% in AP-2 KO mice (Fig. 7b,c, Supplementary Fig. 7v,w). Given the positive feedback loop between TrkB signalling and BDNF expression[47–49], we analysed BDNF mRNA expression levels in WT and AP-2 KO neurons by qPCR (Fig. 7d). Strikingly, BDNF mRNA levels were significantly decreased in the absence of AP-2 compared to WT controls, in agreement with defective TrkB signalling due to stalled autophagosome transport in AP-2 KO neurons. We reasoned that defective arborization of AP-2μ KO neurons should be rescued by boosting BDNF signalling through exogenous application of BDNF. In this setting bath application of BDNF in mass cultures of AP-2 KO neurons is expected to activate their soma-confined TrkB receptors, thereby eliminating the necessity for retrograde transport along the axon. Sustained application of BDNF indeed was sufficient to rescue defective dendritic arborization in the absence of AP-2μ, while it had no effect on the number of branches in WT neurons (Fig. 7e,f),

**Figure 5 | Autophagosome transport by AP-2 promotes neuronal complexity.** (**a,b**) Representative time series (**a**) and corresponding kymographs (**b**) of eGFP-LC3/TrkB-mRFP-positive carriers in the axon. Scale bars, (**a**) 3 μm, (**b**) 5 μm × 5 s. (**c**) Total distance of eGFP-LC3/TrkB-mRFP puncta travelled during a 30 s run when either left untreated or in the presence of BDNF. (**d**) BDNF significantly increases the retrograde velocity of eGFP-LC3/TrkB-mRFP-positive carriers compared to controls (Control: $0.198 \pm 0.0032 \, \mu m \, s^{-1}$, BDNF: $0.315 \pm 0.032 \, \mu m \, s^{-1}$; *$P = 0.036$, $n = 3$ experiments, 16–33 neurites per condition). (**e,f**) WT or AP-2μ KO neurons immunostained for LC3b and phospho-TrkB (pTrkB). White boxes indicate panels magnified to the right. Scale bars, (**e**) 10 μm, (**f**) 2 μm. (**g**) Colocalization of pTrkB with LC3b based on Pearson's coefficient (Rp) (WT: $0.3 \pm 0.02$, AP-2 KO: $0.45 \pm 0.01$, *$P = 0.043$, $n = 4$ experiments, 60–64 ROIs per condition). (**h,i**) Representative epifluorescent images and corresponding kymographs of TrkB-mRFP-positive puncta in WT and AP-2μ KO neurons. Scale bars, 5 μm. (**j**) TrkB velocity is decreased in AP-2 KO neurons compared to WT (WT: $0.39 \pm 0.03 \, \mu m \, s^{-1}$, KO: $0.25 \pm 0.03 \, \mu m \, s^{-1}$, *$P = 0.033$, $n = 3$ experiments, 44–48 neurites per genotype). (**k**) eGFP-expressing WT or AP-2μ KO neurons at DIV14. Scale bar, 40 μm. (**l**) Mean number of branching points in WT (69.78 ± 10.76) and AP-2μ KO neurons (24.25 ± 2.46, *$P = 0.014$, $n = 3$ experiments, 22–25 neurons per genotype). See also Supplementary Fig. 6b. (**m**) Mean number of branching points in WT and AP-2μ KO neurons co-expressing eGFP and AP-2μ (WT + AP-2μ: 144.99 ± 25.31, KO + AP-2μ: 118.38 ± 9.45, $P = 0.308$, $n = 3$ experiments, 29–31 neurons per condition). See also Supplementary Fig. 6a,c. (**n**) Mean number of branching points in control neurons expressing WT HA-AP-2α (HA-AP-2α WT) (144.131 ± 13.43) or LC3-binding deficient mutant AP-2α (HA-AP-2α Mut) (66.36 ± 17.55) (*$P = 0.024$, $n = 4$ experiments, 32–38 neurons per condition). See also Supplementary Fig. 6d,e. (**o**) eGFP-expressing WT or ATG5 KO neurons. Scale bar, 40 μm. (**p**) Mean number of branching points in WT or ATG5 KO neurons (WT: 134.69 ± 16.84; KO: 62.62 ± 18.11) (*$P = 0.013$, $n = 3$ experiments, 35–42 neurons per genotype). See also Supplementary Fig. 6m. Data in **d,g,j,l,m,n,p** are illustrated as box plots as described in Methods. Data in **c** and all data reported in the text are mean ± s.e.m.

in agreement with published data[50,51]. Under similar conditions nerve growth factor (NGF) application failed to rescue the reduced neuronal complexity in AP-2μ-deficient neurons (Fig. 7g).

Taken together our results reveal a novel function for AP-2 in BDNF/TrkB signalling via promoting retrograde transport of

TrkB-containing autophagosomes. As retrograde transport of TrkB-containing endosomes is required for neurotrophins to exert their transcriptional regulation of activity-dependent genes, most notably of BDNF (ref. 49), in the nucleus[38,48], the loss of neuronal complexity and subsequent neurodegeneration in AP-2 KO mice is likely the consequence of defective BDNF gene

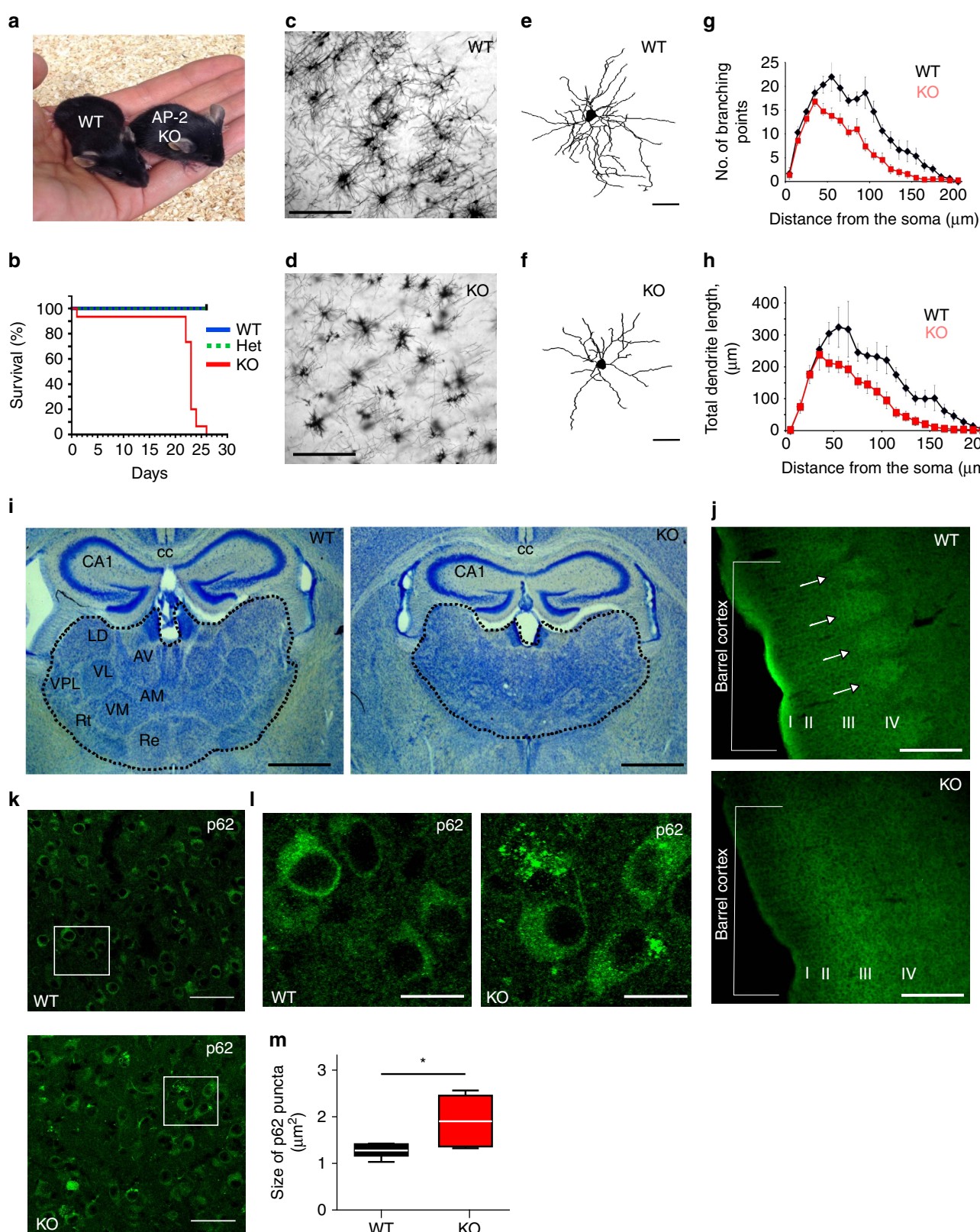

transcriptional regulation due to impaired retrograde autophagosome transport (Fig. 7a).

## Discussion

Our results reveal a novel function of autophagosomes as retrograde shuttles for TrkB-signalling complexes in neurites to promote neuronal branching and to prevent neurodegeneration. We show that TrkB-signalling endosomes, proposed to convey distal BDNF signals to the soma more than 15 years ago[46], may in fact correspond to late-stage LC3/Rab7-positive autophagosomes containing active TrkB complexes[21] (Fig. 7a). In neurons, unlike non-neuronal cells, the formation of autophagosomes and their turnover by fusion with lysosomes is spatially segregated as autophagosomes form in distal axons and mature during their retrograde transport along microtubules to the cell soma, which contains the majority of lysosomes[10]. This arrangement allows neurons to integrate TrkB/BDNF signals from their distal axons to spatiotemporally instruct neuronal branching and to allow for survival via a positive feedback loop that in turn controls BDNF expression, while active TrkB complexes may eventually undergo lysosome-mediated degradation upon their arrival in the neuronal soma. Together with the observation that distal autophagosomes[10] have not yet been acidified and are proteolytically inactive, this predicts that premature acidification of TrkB-containing autophagosomes impairs TrkB/BDNF signalling and, thus, neuronal circuit development (consistent with ref. 18). Indeed, recent data have shown that overacidification of TrkB-containing organelles in the absence of the Christianson syndrome protein NHE6, an $Na^+/H^+$ exchanger known to underlie postnatal microcephaly, intellectual disability, epilepsy and autism[52,53], results in defective neuronal arborization[52] akin to loss of AP-2 or ATG5 reported here.

Retrograde axonal transport of TrkB-containing autophagosomes to the soma is promoted by a protein complex comprising the autophagy protein LC3, the dynein activator p150[Glued] and the endocytic adaptor AP-2 (Fig. 7a), a protein hitherto thought to be exclusively involved in clathrin-mediated endocytosis[24] and reformation of synaptic vesicles[26]. Several lines of evidence suggest that the function of AP-2 in retrograde axonal transport of TrkB-containing autophagosomes is independent of its established role in endocytosis. First, AP-2 is associated and co-trafficked with autophagosomes (Fig. 1) containing active TrkB receptors post-internalization (Fig. 5). Second, and consistent with this, we find AP-2 to be largely dispensable for BDNF-induced TrkB endocytosis in cultured neurons (see Supplementary Fig. 5n–p). Third, we show that defective retrograde transport and accumulation of autophagosomes in neurons in absence of AP-2μ is phenocopied by overexpression of an LC3 binding-deficient mutant version of AP-2α$_A$ that is fully functional with respect to endocytosis (Fig. 4). Finally, we demonstrate that AP-2 via distinct subunits directly associates with LC3 and with p150[Glued] independent of its association with endocytic proteins. The LC3-AP-2-p150[Glued] complex likely is of special importance in neurons, where autophagosomes are transported over large distances and with high precision and speed[10]. A role of AP-2 in neuronal branching is supported by data from mammalian neurons and from Drosophila suggesting a key role for AP-2-associated kinase 1, a crucial activator of AP-2, in dendrite arborization[54]. Interestingly, recent data have identified an endocytosis-independent role for the endocytic protein endophilin in neuronal autophagosome formation[55,56], for example, upstream of the function of AP-2 in autophagosome transport described in this study. How AP-2 switches between its functions in endocytosis and retrograde transport of TrkB-containing autophagosomes remains to be determined.

It is possible, if not likely, that additional factors besides AP-2 act as adaptors for retrograde dynein-based motor complexes to shuttle TrkB-containing autophagosomes to the neuronal soma. These include the Snapin subunit of BLOC-1, which has been postulated to bind to dynein[21] or JIP1, a multifunctional adaptor implicated in both antero- as well as retrograde movement of autophagosomes[32]. These factors could either cooperate with AP-2, for example, to coordinate recruitment of both dynein and p150[Glued]/dynactin, or else distinct retrograde adaptors may operate in distal versus proximal axons. Future studies will be needed to distinguish between these possibilities. Irrespective of the precise molecular mechanisms the identification of autophagosomes as shuttling devices to convey TrkB/BDNF signals to the neuronal soma adds a new facette to the functions of autophagy. Based on our results we predict that autophagy-inducing agents such as polyamines[57] may act as potent therapeutics for the treatment of neurodegenerative and age-associated nervous system disorders, that are often known to be associated with defective axonal transport[58,59], by mediating TrkB/BDNF signalling.

## Methods

**Animals.** Animals were housed in small groups on a 12 h light/dark cycle with food and water ad libitum. All animal experiments were approved by the ethics committees of the LAGeSo Berlin and LANUV Cologne and were conducted according to the committee's guidelines. Wistar rats for the experiments presented in Fig. 2c and Supplementary Fig. 6f–i were obtained from the animal facility of the Mossakowski Medical Center of the Polish Academy of Sciences (Warsaw, Poland). Conditional AP-2 KO (AP-2[lox/lox] × inducible CAG-Cre (ref. 60) and AP-2[lox/lox] × Tubulinα1 Cre) mice were described previously[26]. ATG5[lox/lox] (B6.129S-Atg5[tm1Myok] (RBRC02975)) mice[15] were received from the RIKEN BioResource Center (BRC, Ibaraki, Japan). Conditional ATG5 KO mice were

---

**Figure 6 | Reduced neuronal complexity and neurodegeneration in the absence of neuronal AP-2μ in vivo.** (**a**) Postnatal growth retardation of 21-day-old KO mice conditionally deleted for AP-2μ by transgenic expression of Cre recombinase under the neuron-specific Tα1 tubulin promoter (AP-2[lox/lox]:Tubα1-Cre ). See also Supplementary Figs 7a,b. (**b**) Kaplan–Meier survival curves of neuron-specific AP-2μ KO mice and littermate controls (AP-2[wt/wt]:Tubα1-Cre (WT), AP-2[lox/wt]:Tubα1-Cre (Het) and AP-2[lox/lox]:Tubα1-Cre (KO)). (**c,d**) Golgi silver impregnation of cortices from p20 WT and AP-2μ KO mice reveal the loss of dendritic architecture in AP-2μ KO brain. Scale bars, 200 μm. (**e,f**) 3D morphology of stellate neurons in control (**e**) and AP-2μ-deficient (**f**) brains. Scale bars, 40 μm. (**g,h**) Sholl analysis of stellate cells, revealing their branching complexity (**g**) and total dendritic length (**h**) in p20 WT and AP-2μ KO brains. (**i**) Histopathological analysis of the brain of AP-2μ KO mice at p20 shows marked degeneration of the thalamus (indicated by dotted line), but no overt alteration of the hippocampus (CA1). Nissl-stained brain sections of WT and conditional AP-2μ KO mice. cc, corpus callosum; AV, anteroventral thalamic nucleus; AM, anteromedial thalamic nucleus; LD, laterodorsal thalamic nucleus; VL, ventrolateral thalamic nucleus; VM, ventromedial thalamic nucleus; VPL, ventral posteriorlateral thalamic nucleus, Re, reuniens thalamic nucleus; Rt, reticular nucleus. Scale bars, 800 μm. See also Supplementary Fig. 7g–p for an overview of the temporal progression of neurodegeneration in the brain of AP-2 KO mice. (**j**) Loss of barrel compartments in the somatosensory cortex of AP-2μ KO mice. Cortical barrels visualized by immunostaining for potassium-chloride transporter member 2 are seen in WT controls, but absent in AP-2μ KO brains. Roman numbers indicate cortical layers. Scale bars, 500 μm. (**k,l**) Representative confocal images of thalamus in WT and AP-2μ KO mice immunostained for p62. White rectangular boxes in **k** indicate areas magnified in **l**. Scale bars, 5 μm. (**m**) Mean size of p62-positive puncta is significantly increased in brains of AP-2μ KO mice (1.92 ± 0.26) compared to WT littermates (1.26 ± 0.08, *P = 0.041, n = 5). Data in **m** are illustrated as box plots as described in Methods. Data in **b,g,h** and all data reported in the text are mean ± s.e.m.

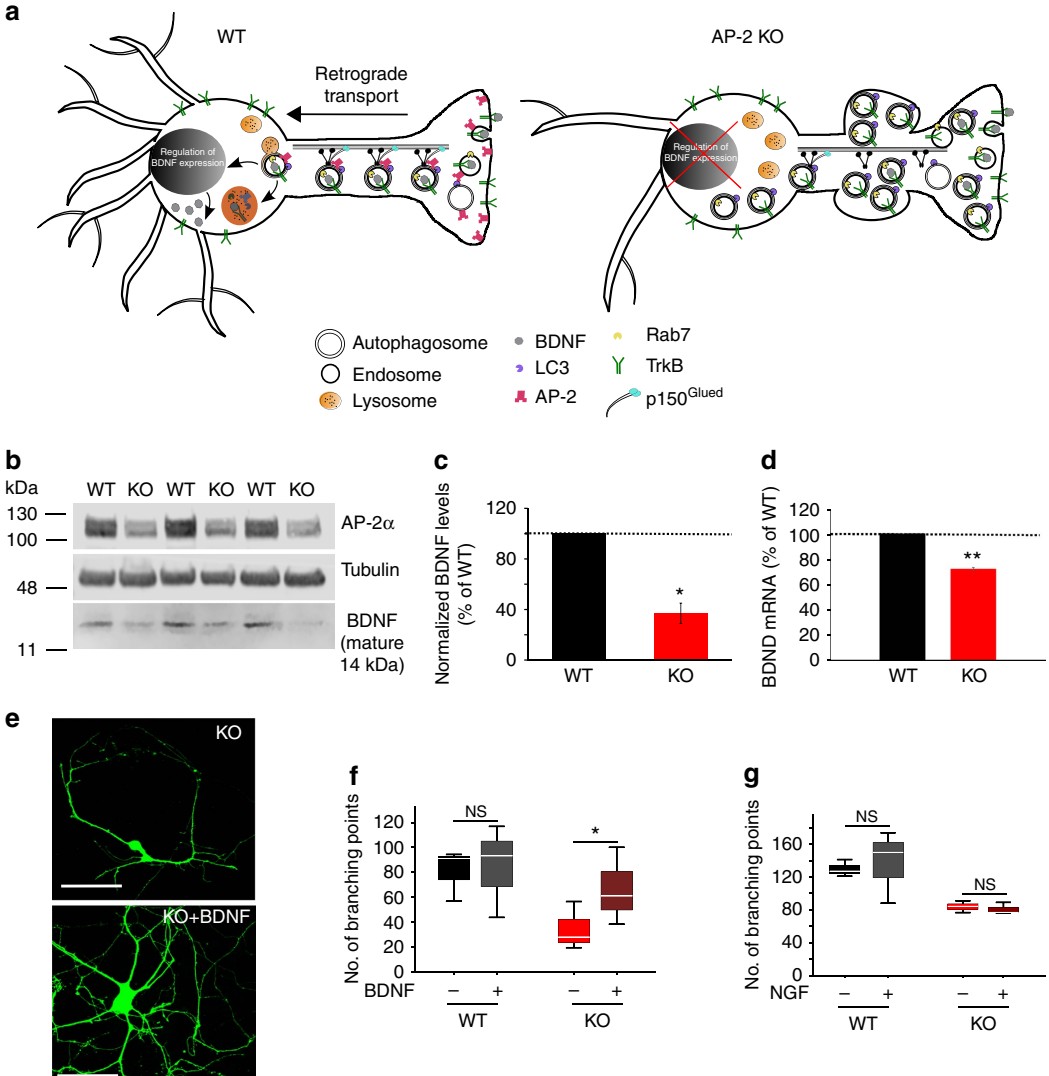

**Figure 7 | Loss of neuronal complexity in AP-2μ-deficient neurons results from reduced TrkB signalling to control expression of BDNF.**
(**a**) Hypothetical model for the role of AP-2 in retrograde transport of TrkB-containing autophagosomes in neurons. In WT neurons, AP-2 via its association with LC3 and p150$^{Glued}$ mediates retrograde transport of BDNF/TrkB-containing amphisomes (late-stage autophagosomes post-fusion with Rab7-positive late endosomes) to the cell body, where TrkB signalling regulates transcription of activity-dependent genes in the nucleus. In the absence of AP-2 (KO) TrkB endocytosis proceeds, however BDNF/TrkB-mediated signalling is defective due to impaired retrograde transport of BDNF/TrkB-containing autophagosomes. Stalled late-satge autophagosomes in neurites of AP-2 KO neurons cause axonal swellings and underlie neurodegeneration.
(**b**) Immunoblot analysis of BDNF expression levels in brain lysates from WT or conditional AP-2μ KO mice. Note that AP-2α remaining in the KO brains is largely derived from AP-2 expressed in glial cells not targeted by Cre. See also the levels of pro-BDNF in WT and AP-2 KO neurons in Supplementary Fig. 7v. (**c**) BDNF levels are significantly reduced in AP-2 KO brains. BNDF levels in WT were set to 100% ($n = 3$ mice per genotype, $*P = 0.0158$). (**d**) BDNF mRNA levels are significantly decreased in AP-2 KO neurons (WT: $99.9 \pm 0.02\%$, KO: $72.1 \pm 1.12\%$, $**P = 0.02$). (**e–g**) Reduced neuronal complexity of AP-2μ-deficient neurons is rescued by long-term BDNF application. (**e**) AP-2μ KO neurons expressing eGFP were treated for 7 days with 50 ng ml$^{-1}$ BDNF or left untreated. Scale bars, 100 μm. (**f**) Application of BDNF rescues the number of branching points in neurons lacking AP-2μ compared to untreated controls (KO + BDNF: $66.65 \pm 24.19$, KO untreated: $34.87 \pm 11.19$, $*P = 0.046$). No difference in the number of branching points between untreated and BDNF-treated WT neurons was observed (WT + BDNF: $84.74 \pm 21.49$, WT untreated $80.86 \pm 11.96$, $P = 0.882$, $n = 3$ experiments, 21–23 neurons per condition). (**g**) NGF fails to rescue the neuronal complexity loss in AP-2μ-KO neurons (KO + NGF: $80.6 \pm 4.2$, KO untreated: $83.8 \pm 4.2$, $P = 0.254$; WT + NGF: $130.1 \pm 6.3$, WT untreated $137.6 \pm 25.4$, $P = 0.774$, $n = 3$ experiments). NS, non-significant. Data in **f,g** are illustrated as box plots as described in Methods. Data in **c,d** and all data in the text are mean ± s.e.m.

created by crossing ATG5$^{lox/lox}$ mice with a Tamoxifen-inducible CAG-Cre line (B6.Cg-Tg(CAG-cre/Esr1*)5Amc/J; the Jackson Laboratory).

**Plasmids.** Expression plasmids encoding HA-tagged AP-2α$_A$ WT or LC3-binding defective mutant AP-2α$_A$ were a generous gift from by Dr P. Greengard (Rockefeller University, New York, USA)[29]. eGFP-mRFP-LC3a was generated from cMyc-LC3a, generously provided by Dr F. Reggiori (University of Utrecht, the Netherlands). mRFP-TrkB was a kind gift from Dr F. Saudou (France Grenoble

Institut des Neurosciences, Grenoble, France). eGFP-TrkB was constructed by placing the coding sequence of TrkB lacking its own signal peptide downstream of the ApoER signal peptide in frame with eGFP. pSUPER (ref. 61), pEGFPC2-BIO, HA-BirA (ref. 62) plasmids were kindly provided by Dr C.C. Hoogenraad (University of Utrecht, the Netherlands). β-actin-GFP and pEGFP-BIO-β-gal (AviGFP-β-gal) were described previously[63,64]. pEGFP-BIO-p150$^{Glued}$ (AviGFP-p150$^{Glued}$) was obtained by cloning rat p150$^{Glued}$ cDNA, obtained by PCR using rat cDNA library, into pEGFPC2-BIO. p150$^{Glued}$ shRNA sequence (5′-gatcgagagacagtcatca-3′) was designed against rat p150$^{Glued}$ mRNA and cloned

into pSuper vector. AP2mu-IRES-mRFP-in-AAV-HBA-EWB was generated by introducing the full-length cDNA encoding mouse µ2-adaptin into the backbone vector AAV-HBA-EWB via Nhe1/Age1 restriction sites. IRES-mRFP obtained by fusion PCR was inserted into the resulting construct via Age1/Bsp1407I restriction sites. Plasmid encoding GST-tagged human LC3b was a gift from Dr Volker Dötsch (University of Frankfurt, Germany). GST-tagged human Stonin 2 (amino acids 1–555) has been described before[65]. mCherry-ATG12 was a generous gift of Dr Michael Lammers (CECAD, University of Cologne). Mito-mCherry was a kind gift of Dr Elena Rugarli (CECAD, University of Cologne), Munc-13-1-eYFP was a gift of Dr Nils Brose (Max Planck for Exp. Medicine, Göttingen, Germany).

**Preparation of neuronal cell cultures and transfection.** Neurons from cortex and hippocampus were isolated from postnatal mice at p1–5 as previously described[66]. Cells were transfected at 7–9 days in vitro (DIV) using an optimized calcium phosphate protocol[66].

To initiate homologous recombination in neurons from floxed animals expressing a tamoxifen-inducible Cre recombinase cultured neurons were treated with 0.25 µM (Z)-4-hydroxytamoxifen (Sigma) immediately after plating (DIV0). Equal concentrations of tamoxifen (0.25 µM) were used during medium renewal on DIV1 and DIV2. Ethanol was added to control neurons in an amount equal to the tamoxifen concentration (0.25 µM). In some cases, homologous recombination was induced at a later stage by treating cultured neurons from floxed animals expressing a tamoxifen-inducible Cre recombinase with 0.25 µM (Z)-4-hydroxytamoxifen (Sigma) at DIV8. For rescue experiments homologous recombination was induced by applying the tamoxifen at DIV0. AP-2µ-mRFP construct was introduced at DIV8 and the neurons were analysed at DIV14.

**Immunocytochemistry and analysis of cultured neurons.** Neurons were fixed on DIV 13–16 in 4% paraformaldehyde (PFA) in phosphate-buffered saline (PBS) for 15 min at RT, washed and permeabilized with 0.3% Triton X-100 (Tx) for 15 min at room temperature. After blocking with PBS containing 10% normal goat serum (NGS) and 0.3% Tx for 1 h, neurons were incubated with primary antibodies (see Supplementary Table 1) for 1 h in PBS containing 10% NGS and 0.3% Tx. Coverslips were rinsed three times with PBS (10 min each) and incubated with corresponding secondary antibodies (see Supplementary Table 1) for 30 min (diluted 1:400 in PBS containing 0.3% Tx and 10% NGS). Subsequently, coverslips were washed three times in PBS and mounted in Immumount. For detection of LC3b and Rab7 permeabilization and blocking were performed in 5% BSA and 0.3% Saponin. The same buffer was used for antibody dilution.

Neurons were imaged at DIV 13–16 at a resolution of 1,024 × 1,024, with eight-bit sampling (no z increment was used) with a Leica SP5 confocal microscope using a Plan-Apochromat × 63/1.32 oil DIC objective, corrected for both chromatic and spherical aberrations. Pearson's correlation coefficients (Rp) were determined in ROIs (11.7 × 13.3 µm) from non-processed raw dual channel images using the Intensity Correlation Analysis function from the Colocalization Macro in ImageJ (ref. 67). For quantitative analysis of fluorescent puncta the total area of the neuron was manually selected using ImageJ selection tools. Fluorescent puncta were determined by applying the autothreshold 'minimum' algorithm implemented in ImageJ and analysed using the 'Analyze particles' ImageJ module to determine the number of fluorescent puncta per µm$^2$. All antibodies used for immunostaining are indicated in Supplementary Table 1 (that is, IF).

**STED imaging and analysis.** STED imaging with time-gated detection was performed using a commercial Leica SP8 TCS STED microscope (Leica Microsystems) equipped with a pulsed excitation white light laser (WLL; ∼80-ps pulse width, 80-MHz repetition rate; NKT Photonics) and two STED lasers for depletion (continuous wave at 592 nm, pulsed at 775 nm). The pulsed 775-nm STED laser was triggered by the WLL. Within each independent experiment, samples were acquired with equal settings. Alexa 488 and Alexa 532 were excited using a pulsed WLL at 488 and 545 nm, respectively. Depletion occurred at 592 nm. Fluorescence signals were detected sequentially by hybrid detectors at appropriate spectral regions separated from the STED laser by corresponding dichroic filters. Images were acquired with an HC PL APO CS2 × 100/1.40-N.A. oil objective (Leica Microsystems), a scanning format of 1,024 × 1,024, eight-bit sampling, and 4.5 zoom, yielding a pixel dimension of 25.25 nm and 25.25 nm in the x and y dimensions, respectively. Pearson's correlation coefficients were determined in ROIs (5.1 × 4.3 µm) using the Intensity Correlation Analysis function from the Colocalization Macro in ImageJ (ref. 67).

**Sholl analysis of cultured neurons.** Neurons were transfected with eGFP on DIV 8. In case the cells were treated with growth factors (BDNF and NGF, both Peprotech), these factors were added on DIV 7 at 50 ng ml$^{-1}$. Cells were either fixed on DIV 14/15 for AP-2µ KO and control neurons, or on DIV20-22 for ATG5 KO and control neurons and processed as above. In case homologous recombination was induced at DIV8, AP-2 KO neurons were fixed on DIV19-20. Neurons were imaged with a Leica SP5 confocal microscope using a Plan-Apochromat × 63/1.32 oil DIC objective, corrected for both chromatic and spherical aberrations. Neurons were scanned at a resolution of 1,024 × 1,024, eight-bit sampling, zoom 1 and z-increment of 0.5 µm. Sholl analysis of single cells

was performed using the ImageJ Sholl Analysis Macro. The complexity of neuronal branching was calculated by summing the number of intersections within 200 µm from the cell body.

**Live imaging of cultured neurons.** Neurons were imaged on 12–16 DIV using an inverted Zeiss Axiovert 200M microscope with a × 40 oil-immersion objective, temperature-controlled stage (37 °C) and a CCD camera. Time-lapse images of neurons expressing TrkB-mRFP and/or mRFP-eGFP-LC3 were acquired every second using Slidebook (Improvision Germany, Göttingen) for 30 s. For starvation experiments, neurons were kept in serum-free media for 3 h before imaging. Individual particles were manually identified in Slidebook. Particle tracking analysis of individual particles was done using the Particle tracking protocol in Slidebook, and individual tracks obtained were verified manually. The mean speed of individual particles was calculated by determining the mean square displacement between each pair of sequential frames averaged over all frames. Kymographs were generated using the corresponding module in Slidebook. Frequency distribution analysis was done in Excel (Microsoft).

**Immunohistochemical analysis of brain sections.** Mice were killed on postnatal days 3, 7, 14 or 18–21 by an overdose of ketamin/rompun (i.p., 10 µl per 10 g body weight) and transcardially perfused with 50 ml saline solution (0.85% NaCl, 0.025% KCl, 0.02% NaHCO3, pH 6.9, 0.01% heparin, body temperature), followed by 50 ml cold (7–15 °C) freshly depolymerized 4% (wt/vol) PFA (Merck) in 0.1 M PBS, pH 7.4. Brains were carefully taken out of the skull, postfixed overnight in the same fixative, and placed in a mixture of 20% (vol/vol) glycerol and 2% (vol/vol) dimethylsulfoxide (VWR International) in 0.4 M PBS for 24 h for cryoprotection. Frozen horizontal, coronal or sagittal sections (40 µm) were collected in six series in 0.1 M PBS. For immunostaining, corresponding hippocampal sections from WT and KO littermates were processed simultaneously. Sections from the second series were washed three times in PBS (3 × 15 min each), followed by washing several times in PBS containing 0.3% Triton X-100 (9 × 20 min each). Sections were preincubated with PBS containing 5% (vol/vol) NGS and 0.3% Triton X-100 for 1 h and subsequently incubated with primary antibody at 4 °C for 48 h (see Supplementary Table 1 for a list of antibodies). Then, sections were washed nine times for 20 min each in 0.3% Triton X-100 in 0.1 M PBS and incubated with Alexa-conjugated secondary antibodies (1:400) for 12 h using standard techniques. Finally, sections were mounted on gelatin-coated glass slides. Sections were imaged at a resolution of 1,024 × 1,024, with eight-bit sampling (no z increment was used) with a Leica SP5 confocal microscope using a Plan-Apochromat × 63/1.32 oil DIC objective, corrected for both chromatic and spherical aberrations. Fluorescent puncta were determined in the background-substracted ROI (90 × 83 µm) by using the autothreshold 'Max Enthropy' algorithm implemented in ImageJ and analysed using the 'Analyze particles' ImageJ module to determine the number of fluorescent puncta per µm$^2$. All antibodies used for immunostaining are indicated in Supplementary Table 1 (under IF).

**Nissl staining.** Sections were mounted in 0.2% gelatine solution in 250 mM Tris-HCl and dried overnight at 40 °C on a heating plate. Mounted sections were re-hydrated for 1 min in water and stained for 5–10 min in 0.1% cresyl violet solution. Subsequently, sections were rinsed three times (2 min each) in water and dehydrated using an ascending ethanol series (50, 70, 80, 90%) for 2 min each. After rinsing the sections in 96% ethanol, they were destained with 0.5% acetic acid and washed twice in 100% ethanol (2 min each), incubated with xylene for 2 min and subsequently mounted using Entellan.

**Golgi silver impregnation of neurons in vivo.** Golgi silver impregnation was performed on 200 µm thick perfused brain sections from p20 mice using the FD Rapid Golgi Stain Kit (FD Technologies). Cells were reconstructed and analysed using Neurolucida (MBF Bioscience).

**Fluoro-Jade C staining of brains.** Fluoro-Jade C (FJC) a polyanionic fluorescein derivative that can sensitively and selectively bind to degenerative neurons[68]. To visualize degenerating neurons, brain sections from perfused p20 WT and AP-2 KO brains were mounted of gelatin-coated slides and subjected to FJC staining, according to manufacturer's protocol (Fluoro-Jade C staining Kit, Biosensis). Sections were imaged with a Leica SP5 confocal microscope equipped with FITC filter.

**TrkB endocytosis assay in cultured neurons.** Hippocampal WT and AP-2 KO neurons were prepared as described above and transfected on DIV 8 with eGFP-TrkB. On DIV 13 coverslips were rinsed once in 1 ml osmolarity-adjusted serum-free NBA-containing medium. To induce antibody uptake neurons were incubated for 15 min or, alternatively, for 30 min at 37 °C in pre-warmed 70 µl serum-free medium supplemented with 50 ng ml$^{-1}$ human BDNF (Peprotech) and 1:500 diluted GFP-antibody (polyclonal rabbit, Abcam, ab6556). Control condition corresponded to the incubation for 30 min at 4 °C in pre-cooled antibody/BDNF solution. After the indicated time points coverslips were transferred to a new well

plate containing ice-cold PBS and washed once for 1 min with ice-cold 0.2 M acetic acid and 0.5 M NaCl in MQ. After two additional on/off washing steps with ice-cold PBS neurons were fixed for 10 min at RT with 4% PFA/4% sucrose and washed three times with PBS. Subsequently cells were blocked in PBS supplemented with 10% NGS (blocking buffer) under non-permeabilizing conditions for 30 min. The surface-bound antibody pool was immunostained by incubation with goat-anti-rabbit Alexa Fluor 647 coupled antibody (Invitrogen, 1:100 in blocking buffer) for 1 h at room temperature. Subsequently, the cells were washed three times in PBS and postfixed for 10 min at RT with 2% PFA/2% sucrose followed by three PBS washes. To identify the internalized antibody pool the neurons were permeabilized, blocked in the blocking buffer supplemented with 0.3% TX-100 for 20 min and immunostained with goat anti rabbit Alexa Fluor 568 labelled secondary antibodies (1:200, blocking buffer supplemented with 0.3% Tx) for 30 min at RT. Coverslips were washed three times with PBS and mounted using Immumount. Cells were imaged using a Leica SP8 and analysed using ImageJ. The amount of internalized eGFP-TrkB receptors was calculated by taking the ratio between the mean grey value of internalized GFP antibody over the mean grey value of total expression levels of eGFP-TrkB.

**HeLa cell siRNA and plasmid transfections.** HeLa cells were subjected to two rounds of transfection (days 0 and 2) with siRNA using Oligofectamin (Life Technologies) according to the manufacturer's protocol. For additional transient overexpression, plasmids were transfected on day 4 using X-tremeGENE 9 DNA transfection reagent (Roche). For silencing, the following siRNA was used: AP-2α 5′- AAGAGCAUGUGCACGCUGGCCA. Scrambled AP-2μ sequence 5′-GTAAC TGTCGGCTCGTGGTTT was used as control siRNA.

**Transferrin uptake in HeLa cells.** HeLa cells seeded on coverslips coated with Matrigel (BD Biosciences) were serum-starved for 1 h and treated with 25 μg ml$^{-1}$ Tf-Alexa647 (Life Technologies) for 10 min at 37 °C. Cells were washed twice with ice-cold PBS and acid washed at pH 5.3 (0.1 M Na-acetate, 0.2 M NaCl) for 1 min on ice. The coverslips were washed twice with ice-cold PBS and fixed with 4% PFA for 30 min at room temperature. Transferrin uptake was analysed using a Nikon Eclipse Ti microscope equipped with a × 40 oil-immersion objective and quantified using ImageJ software.

**Immunoblot analysis of mouse brain extracts.** Somatosensory cortices were homogenized in lysis buffer (20 mM Hepes-KOH, pH 7.4, 100 mM KCl, 2 mM MgCl$_2$, 1% Triton X-100, supplemented with 1 mM PMSF and mammalian protease and phosphatase inhibitor mixture) using a glass teflon homogenizer. The lysate was incubated 20 min on ice before centrifugation at 17,000$g$ for 10 min at 4 °C. The protein concentration was determined by Bradford assay. Samples were analysed by SDS–PAGE and immunoblotting. Proteins were detected using HRP-coupled secondary antibodies in a chemo-luminescence reaction or with the Odyssey LiCor system. All antibodies used for immunoblotting are indicated in Supplementary Table 1 (under WB). Images in Fig. 7b, supplementary Figs 3c, 4g, h, 5g, i, l, 6f, 7e have been cropped for presentation. Full size images are presented in supplementary Fig. 8.

**Immunoprecipitation.** Dynabeads Protein G (50 μl) were washed in lysis buffer (150 mM NaCl, 1 mM CaCl$_2$, 1 mM MgCl$_2$, 20 mM HEPES-NaOH, pH 7.5) supplemented with 0.3% CHAPS and protease, as well as phosphatase inhibitors. Beads were incubated with 10 μg antibody for 15 min at room temperature, and unbound antibody was then removed by washing. Rat brain lysates were prepared by homogenization with a glass-teflon homogenizer with 30 strokes in lysis buffer (10 × volume/weight) in the absence of detergent. For tissue lysis the homogenate was mixed at a 1:1 ratio with lysis buffer containing 0.6% CHAPS and rotated for 1 h at 4 °C. Cell debris was removed and the protein concentration adjusted to 3 mg ml$^{-1}$ by addition of lysis buffer containing 0.3% CHAPS. lysate (700 μl) were incubated over night at 4 °C with freshly prepared Dynabeads. Samples were washed three times using 300 mM NaCl, 1 mM CaCl$_2$, 1 mM MgCl$_2$, 0.1% CHAPS, 20 mM HEPES-NaOH, pH 7.5. Proteins were eluted in SDS sample buffer and analysed via SDS–PAGE and immunoblotting. All antibodies used for immunoprecipitation experiments are indicated in Supplementary Table 1 (under IP). Images in Fig. 2c have been cropped for presentation. Full size images are presented in Supplementary Fig. 8.

**Binding of biotinylated p150$^{Glued}$ to the β2-ear.** M-280 streptavidin Dynabeads were blocked for 1 h in 20 mM HEPES-KOH, pH 7.5, 150 mM KCl supplemented with 0.2 mg ml$^{-1}$ BSA and 20% glycerol. Beads were washed with lysis buffer (20 mM HEPES-KOH, pH 7.5, 150 mM KCl, 0.3% CHAPS) supplemented with protease- and phosphatase inhibitors. HEK293T cells co-expressing AviGFP-p150$^{Glued}$ or AviGFP-β-galactosidase as a negative control were lysed for 15 min on ice in lysis buffer. The supernatant was added to the prepared M-280 streptavidin Dynabeads for 1 h at 4 °C while rotating. Beads were then washed four times in wash buffer (20 mM HEPES, pH 7.5, 500 mM KCl, 0.1% CHAPS) and incubated with 0.2 mg ml$^{-1}$ His$_6$-AP-2β appendage domain in binding buffer: 20 mM HEPES-pH 7.5, 300 mM KCl and 0.1% CHAPS for 2 h at 4 °C rotating.

Finally, beads were washed four times with binding buffer, eluted in SDS sample buffer and analysed via SDS–PAGE and subsequent immunoblotting. Images in Fig. 2d have been cropped for presentation. Full size images are presented in Supplementary Fig. 8.

**Brain lysate pull-down assay.** GST- and His-tagged fusion proteins were affinity-purified by glutathione-sepharose respectively Ni-NTA based affinity chromatography. GST-fusion-proteins were stored at 4 °C in PBS. Detergent extracted mouse brain lysates were prepared as described[66]. Briefly, mouse brains were homogenized in lysis buffer (10 mM HEPES, pH 7.4, 150 mM NaCl, 2 mM DTT, 0.05% Triton X-100, supplemented with protease- and phosphatase inhibitors) and incubated on ice for 30 min before centrifugation at 65,000 × r.p.m. for 15 min at 4 °C. Supernatant was added to 50 μg recombinant purified GST or GST-AP-2a$_{A/C}$ and rotated for 2 h at 4 °C. Samples were washed five times with lysis buffer, eluted in SDS sample buffer and analysed via SDS–PAGE and immunoblotting. Images in Fig. 2b,e and Supplementary Fig. 2b,c have been cropped for presentation. Full size images are presented in Supplementary Fig. 8.

**Direct binding of LC3b to the AP-2α -appendage domains.** Recombinant purified GST or GST-AP-2a$_{A/C}$-appendage domains (10 μg) and His$_6$-LC3b (5 μg) were incubated in binding buffer (150 mM NaCl, 10 mM HEPES, pH 7.5, 5 mM imidazole, 2 mM DTT and 0.005% Triton X-100) for 1 h at 4 °C on a rotating wheel. Samples were washed five times in binding buffer, proteins were eluted in sample buffer and analysed via SDS–PAGE and immunoblotting for LC3b. Images in Fig. 2a have been cropped for presentation. Full size images are presented in Supplementary Fig. 8.

**Electron microscopy.** Neurons were immersed into PBS buffered 2% glutaraldehyde, followed by 1% OsO$_4$, 1,5% potassium hexacyanoferrat postfixation and uranyl acetate staining. Afterwards neurons were dehydrated in methanol gradient and propylene oxide, followed by Epoxy resin infiltration. After Epoxy resin polymerisation, coverslips were removed from flat embedded cultures with liquid nitrogen assistance. Fifty nanometres sections were cut, contrasted with uranyl acetate and lead citrate and analysed at Zeiss 900 transmission electron microscope. Images were obtained with Olympus MegaViewIII camera or Olympus Morada G2 digital cameras at × 30,000 magnification.

**Quantitative real-time RT – PCR (qRT–PCR).** Total RNA from cultured ethanol or tamoxifen-treated neurons from AP-2μ$^{lox/lox}$:CAG-Cre mice was extracted using TRIzol reagent (ambion, RNA, Life Technologies). After homogenizing the samples with TRIzol reagent, chloroform was added and the aqueous layer containing the RNA was isolated and precipitated with isopropanol. Reverse transcription into cDNA was carried out by using the High Capacity cDNA Reverse Transcription Kit (4368814, Thermo Scientific) according to manufacturers' instructions. qRT–PCR was performed with a LightCycler 1.5 instrument using Fast Start DNA Master SYBR Green I Kit (Roche). Primer sequences are listed in Supplementary Table 2. Samples were run in duplicates and BDNF signals were normalized to GAPDH intensity.

**Statistical analysis.** For analysis of experiments, statistically significant estimates were obtained from independent experiments ($n$). The statistical significance between two groups for all normally distributed raw data except growth factor treatments was evaluated with a two-tailed unpaired student's $t$-test. Effects of BDNF and NGF on neuronal complexity were evaluated using paired student's $t$-tests. The statistical significance between more than two groups for all normally distributed raw data (Fig. 4b,f) was evaluated using one-way ANOVA (Tukey $post-hoc$ test was used to determine the statistical significance between the groups). All normalized data (Figs 4h and 7c,d, Supplementary Figs 3d, 4i, 5h,j,m) were evaluated using one-sample student's $t$-test. Significant differences were accepted at $P < 0.05$. For box plots the median divides the box, while the upper boundary of the box corresponds to the third quartile and the lower boundary corresponds to the first quartile. The minimum and the maximum values extend as bars from the bottom and top of the box.

**Antibodies.** An overview of all antibodies used in this study is given in Supplementary Table 1.

**Data availability.** The data that support the findings of this study are available from the corresponding authors on reasonable request.

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

## Acknowledgements

We thank Sabine Hahn, Delia Löwe, Nina Ellrich and Silke Zillmann for expert technical assistance. Supported by grants from the German Research Foundation (SFB958/A01 to V.H. and T.M.), the NeuroCure Cluster of Excellence (Exc-257) and the Reinhart-Koselleck Award (to V.H.). Work by N.L.K. was supported by a NeuroCure Female PostDoc fellowship and financed by the Cologne Excellence Cluster on Cellular Stress Responses in Aging-Associated Diseases (CECAD, Exc 229). Work of J.J., A.T., A.R.M., A.S. was financed by a National Science Centre grant no. 2011/03/B/NZ3/01970. J.J. and A.R.M. are also recipients of the Foundation for Polish Science 'Mistrz' Professorial Subsidy and Fellowship, respectively. M.K. received funding from an H2020 EU Marie Curie IF fellowship. Furthermore, we thank Prof. Paul Greengard (Rockefeller University, New York, USA) for providing plasmids for WT and LC3-binding defective mutant AP-2α$_{A,}$ Dr Fulvio Reggiori (University of Utrecht, Netherlands) for the Myc-LC3 plasmid, Prof Volker Dötsch (University of Frankfurt, Germany) for the GST-LC3b construct, Dr F. Saudou (France Grenoble Institut des Neurosciences, Grenoble, France) for the TrkB-mRFP plasmid, Dr N. Brose (Max Planck for Experimantal Medicine, Göttingen, Germany) for the Munc-13-1-eYFP plasmid, Dr Eelena Rugarli (CECAD, University of Cologne) for the Mito-mCherry plasmid and Dr Lammers (CECAD, University of Cologne, Germany) for the mCherry-ATG12 plasmid. We are grateful to Dr Min Kye for the generous help with qPCR experiments.

We appreciate the help of the CECAD Imaging Facility, especially Dr Christian Jüngst for the assistance in scanning histology slides.

## Author contributions

N.L.K., G.A.C., M.K., D.P., A.R.M., S.B., A.T., A.S. and J.J. performed experiments, T.M. contributed reagents, N.L.K., J.J. and V.H., designed research. N.L.K. and V.H. wrote the manuscript with input from all authors.

## Additional information

**Competing interests:** The authors declare no competing financial interests.

