## [Peer Review File · Nature Communications]

Reviewers' comments:

Reviewer #1 (Remarks to the Author):

This paper by Kononenko et al identifies a new, non-canonical, role for the adaptor protein AP2 in neuronal function. This is novel because to date AP2 was thought to function only in the endocytic retrieval of synaptic vesicles but this paper shows that AP2, potentially via the α A isoform is also important for the retrograde trafficking of TrkB/BDNF signals to the cell soma during autophagy. Using a conditional knock-out AP2 μ -subunit mouse line, the authors show that retrograde transport of autophagosomes is inhibited and that there is accumulation of autophagosomes and autophagosome components distally in the cultures. With biochemistry they show that AP2 associates with components of the autophagosomes and by cell imaging show that AP2 mediates retrograde transport of BDNF/activated TrkB. They also show a lack of neuronal complexity in the AP2 knock out neurons as well as other conditions where autophagosome maturation and trafficking is inhibited, suggestive of a role for autophagy in neuronal complexity. This could also be rescued by treatment with BDNF, linking the defective retrograde autophagy transport to a functional neuronal read-out.

I think that this work is presented to a high standard and makes valid conclusions on the whole. To my knowledge the statistical analysis presented was valid and in general there is a reasonable level of detail regarding the experimental protocols presented. There were, however, a few cases where figures are mis-referenced (for eg first paragraph results refers to sup fig 1 d-j which doesn't exist and in figure 3 the legend and panel label do not match up).

I feel that the work would benefit, however, from the following points being addressed:

1) I am convinced of the interaction data displayed in figure 2 but I feel that some more points need to be addressed with it. Could the authors please clarify how many experiments these figures represent since there is no indication of the n number in the legend? There are also some controls missing which I feel would enhance the interpretation of the figure for example, including GST loading controls for completeness for the blots shown in a, b and e would rule out differences in fusion protein input being responsible for the difference in binding. It is important for panel a since the binding to the AP2 α C isoform may not be any higher than the GST alone, which considering the authors suggest a potential regulatory mechanism involving specific isoforms for different roles, it is important to establish. Since the authors have the GST-AP2 constructs, it would also be good to see the reverse pulldown of fig 2b with AP2 as bait, looking for LC3b. Further to that, does the AP2 co-IP any of the subunits for anterograde trafficking – potentially would expect not based on the preference of retrograde trafficking but could strengthen that argument.

2) Since the authors postulate that this new role for AP2 may be specific to the AP2 α A isoform, does overexpression of the C isoform fail to rescue the retrograde velocity or the branching phenotype for example? Does it also have the LIR binding site shown to be important in the A isoform?

3) Point for discussion perhaps is the loss of neuronal complexity shown by AP2 KO. The authors show a similar phenotype with ATG5 KO which is involved in the autophagosome maturation pathway but this does not mean that loss of AP2 causes the same phenotype via this pathway. How does this compare to KO of other endocytosis proteins for eg., can the lack of arborisation also be due in part to AP2's role in endocytosis?

4) Given the well known role of autophagosomes in neurodegeneration, I can understand the authors argument that AP2 KO morphological brain defects seen are attributed to neurodegeneration but I do not feel that the authors have completely excluded the possibility that the defects were caused by a defect in development rather than specifically neurodegeneration. The mouse is missing AP2 which does have essential roles in endocytosis and so arguably could also be responsible for the defects. Were the structures ever there in the first place given that the mice are described as lagging behind in

development from birth? I think that this distinction is important since if the structures were not correctly formed, it is not the same as the structures degenerating. Do the authors have any further evidence to support their argument? It does not make the trafficking conclusions of the paper any less valid or interesting but it may require the neurodegeneration conclusions to be a little less definitive.

Minor points:

1) Figure 1: the authors measure the colocalisation of the eGFP-LC3 and mRFP-AP2 with the Pearson's coefficient, but what is the R_p of the empty control vector i.e. mRFP for comparison? Likewise, a +ve interactor for LC3 may also be helpful to get a feel for the extent of colocalisation.

2) The authors also show colocalisation with an autophagosome marker, but is there any colocalisation with other things like motor proteins since biochemically there is evidence of binding to motor proteins? Related to that, is there colocalisation with synaptic markers? I am not disputing that the puncta traffic, but I was interested in the stationary ones, are they at synapses?

3) Several of the quantification graphs where conditions have been normalised prior to plotting, are described in the legends as normalised to 100 % but graph shows normalised to 1. I appreciate it is essentially the same thing but would be easier to understand if they were consistent.

4) Characterisation of endocytic proteins in the KO cultures is only displayed for AP2a levels. The remainder of the graphs is of the p21 mouse lysates and this distinction is not clear in the text. The two may be different due to the fact that the mouse developed to p21 without AP2 but the cultures are made from mice which develop normally with AP2 and then the cultures are treated to delete the AP2. Culture lysates could be blotted for the other endocytic proteins for best comparison or at least, clear distinction made in the text.

5) Could the authors also comment on the anterograde velocity in the KO cells expressing mRFP-eGFP-LC3 since the text describes bidirectional movement? The title of this section focuses on conclusions about retrograde transport but the authors have not excluded an effect on anterograde trafficking.

6) The authors conclude that retrograde transport is reduced in KO neurons by illustrating that the velocity is reduced. Are there the same number of autophagosome puncta moving? I.e. are the proportions of stationary vs retrograde and anterograde similar?

7) Folimycin application experiments showing accumulation of LC3b puncta. How does application of folimycin on wt cells compare to AP2 KO cells? Data is shown for these conditions but not together so hard to compare to what extent KO accumulates LC3b puncta in the context of treatments known to cause accumulation of autophagosomes. A side by side comparison would be helpful or the puncta/ μm^2 values instead of/in addition to the normalised data in panel m.

8) The text states that mTORC1 signalling was "slightly upregulated" (results AP2 regulates autophagosome transport and turnover independent of its role in endocytosis line 11). Since the quantification does not show any statistical upregulation, this statement should be toned down a little to reflect a possible trend since the only component that is upregulated is the inactive form of the S6 kinase. Related, when describing the rescue by mutant AP2 in fig 4, the statistics displayed do not show that the mutant is better than the wild type.

9) The authors look at accumulation of pTrkB in neurites, what happens to non-phosphorylated TrkB or another receptor such as p75 which is activated by BDNF but does not couple to this pathway? Is this accumulation specific to the activated form or a consequence of a more generalised stalling of trafficking and recycling?

10) Does treatment with BDNF do anything to the retrograde velocity in KO neurons (fig 5)? Arguably probably not since there is no AP2 in the KO but given the experiments that come later with rescued branching, it is important to know.

Reviewer #2 (Remarks to the Author):

The manuscript "Retrograde transport of TrkB-containing autophagosomes via the endocytic adaptor AP-2 mediates neuronal complexity and prevents neurodegeneration" by Kononenko et al describes a novel role for the well-characterised adaptor protein AP-2 in dynein-dependent transport of autophagosomes. This is a rather unexpected finding that is well supported by the large amounts of data presented. I have some queries/ comments for this section:

Can the authors comment on clathrin recruitment to these autophagosomes? What percentage of autophagosomes is AP-2 positive, and would this suggest a PIP2 enrichment in the autophagosome membrane of these autophagosomes? Conversely, seeing that the authors observe bidirectional movement of AP-2 positive carriers, but preferentially retrograde movement of autophagosomes, could they speculate on the nature of anterograde AP-2 positive carriers? And is there any difference in the total number of autophagosomes in absence vs presence of AP-2?

A minor comment: Could you please move the explanation of folimycin function from p7 to p6 (where it is first mentioned)?

The authors then go on to discuss that autophagosomes traffic TrkB, and this process is required for BDNF-TrkB signalling. They further state that disruption of this process leads to neurodegeneration. This section would benefit from more detailed descriptions and/or experimental timelines.

The authors state that BDNF treatment leads to an increase in the speed RFP-TrkB movement (is this an increase in speed or in % moving vesicles?). What time after treatment is this seen? Also, overexpression of RTKs frequently leads to auto-activation – could the authors comment please? Also, the kymographs suggest that AP-2 ablation stops TrkB movement in both directions? Is other transport (eg mitochondria) intact?

The authors go on to show reduced dendritic complexity (have they looked at axons?) in the AP-2 KO, which can be rescued by re-expression. It appears to me that the rescue leads to higher complexity than wildtype? Therefore AP-2 overexpression drives dendritic complexity? Please comment. Also, please give experimental details/timelines on when cre was activated and when the rescue constructs were transfected. Is this a developmental effect or pruning? What DIV are the cultures?

Similar for the in vivo data: I presume this is a snapshot of one time point (seeing how rapidly the KOs succumb). At what age are dendritic complexity and gross brain morphology assessed?

The claim that this is "degeneration" (eg manuscript title; p11) would need further time points to show reduction over time – seeing the rather young age at which the mice die this might be a developmental defect? Please clarify or add time points.

Finally, the authors show reduced levels of BDNF in the AP-2 KO mice, and conclude that the morphological defects are due to lack of positive feedback. I am a bit lost here. Have the authors looked at the induction of other IEGs downstream of BDNF/TrkB (eg Arc)? How does replenishing BDNF lead to rescue – what is downstream? And how is retrograde transport required in mass culture (fig 5), but not if exogenous BDNF is added? Also, it is rather curious that the authors do not find an otherwise well-documented effect of BDNF on neuronal complexity in wildtype neurons (eg work of McAllister, Cohen-Cory etc). Please explain. It would be nice to see the full BDNF blot (or ELISA) in figure 7b to assess whether it is BDNF levels or processing that is defective.

Also, how does a receptor signal once within the autophagosomal double membrane structure?

Reviewer #3 (Remarks to the Author):

In this manuscript, Kononenko et al. investigated a novel mechanism of transport of TrkB-containing autophagosomes via the endocytic adaptor protein AP-2. They used biochemical, cell based and in vivo studies to provide largely strong evidence for this mechanism. What is less compelling is the

relevance to neurodegeneration as the deficits they observe could also be developmental.

Specific comments:

Fig. 3, Panel d: EM shows enlarged dense vesicular bodies with concentric multilamellar structures. These could be lysosomes or late stage amphisomes. The cigar-shaped structure in the KO image which looks like a mitochondrion also seems enlarged and dense. Is the magnification the same? Please provide additional images. How do the lysosomes in the cell soma (around the nucleus) look like in wild-type and AP2 KO neurons?

Fig. 3, Panel m: treatment with folimycin caused equal accumulation of LC3b-positive puncta in AP2 KO neurons compared to controls. However, if the transport to the lysosome is delayed in AP2 KO neurons, one might expect to see less accumulation of LC3b/Rab7 puncta over the duration of the experiment. Please clarify. In this regard, why did the authors measure starvation-induced flux in Fig 5, panel b and panel f rather than flux with and without a lysosomotropic inhibitor? How long was the starvation? I was unable to find this information.

Fig. 3, Panel n is missing from the figure

Please include Rab5 (early endosome marker) as an additional control for immunofluorescence to show whether early endosomes change in AP2 KO mice?

Fig 6: The data presented here are interesting but do not distinguish between a developmental and truly degenerative phenotype. Therefore caution is required when implicating this pathway in neurodegeneration.

The accumulation of p62 puncta both in vivo and in cells suggest a broader defect in autophagic degradation. How do the authors explain this if AP2 is more selectively involved in the pathway they describe?

BDNF addition accelerates the transport of LC3/TrkB carriers. However, the data linking LC3/TrkB trafficking to BDNF transport and its nuclear signalling is weak and could be strengthened by studies using (1) labeled BDNF in WT and AP2 KO neurons and (2) addition of BDNF in WT and AP KO neurons cultured in microfluidic devices where one would expect that the KO neurons would not respond to BDNF added at the synaptic terminal compartment.

Detailed response to the reviewers

(reviewers' comments are given in italics, our response to each point is given below).

We would like to thank all three referees for their careful reading of our Ms and for their supportive and constructive comments that have greatly helped in improving our study and tailoring the paper for the readership of *Nature Communications*.

Reviewer #1:

This paper by Kononenko et al identifies a new, non-canonical, role for the adaptor protein AP2 in neuronal function. This is novel because to date AP2 was thought to function only in the endocytic retrieval of synaptic vesicles but this paper shows that AP2, potentially via the αA isoform is also important for the retrograde trafficking of TrkB/BDNF signals to the cell soma during autophagy. Using a conditional knock-out AP2 μ -subunit mouse line, the authors show that retrograde transport of autophagosomes is inhibited and that there is accumulation of autophagosomes and autophagosome components distally in the cultures. With biochemistry they show that AP2 associates with components of the autophagosomes and by cell imaging show that AP2 mediates retrograde transport of BDNF/activated TrkB. They also show a lack of neuronal complexity in the AP2 knock out neurons as well as other conditions where autophagosome maturation and trafficking is inhibited, suggestive of a role for autophagy in neuronal complexity. This could also be rescued by treatment with BDNF, linking the defective retrograde autophagy transport to a functional neuronal read-out.

I think that this work is presented to a high standard and makes valid conclusions on the whole. To my knowledge the statistical analysis presented was valid and in general there is a reasonable level of detail regarding the experimental protocols presented.

Response: We thank the referee for these very positive remarks and for highlighting the importance and high quality of our work.

There were, however, a few cases where figures are mis-referenced (for eg first paragraph results refers to sup fig 1 d-j which doesn't exist and in figure 3 the legend and panel label do not match up).

Response: We thank the referee for alerting us to these errors that have been corrected in the revised version of our Ms.

I feel that the work would benefit, however, from the following points being addressed:

1) I am convinced of the interaction data displayed in figure 2 but I feel that some more points need to be addressed with it. Could the authors please clarify how many experiments these figures represent since there is no indication of the n number in the legend?

Response: We have added the n number of independent experiments that were conducted for each panel in figure 2. Many of these experiments have been conducted even more often with slight variations in the conditions, yet, similar result that we have not counted.

There are also some controls missing which I feel would enhance the interpretation of the figure for example, including GST loading controls for completeness for the blots shown in a, b and e would rule out differences in fusion protein input being responsible for the difference in binding. It is important for panel a since the binding to the AP2 αC isoform may not be any higher than the GST alone, which considering the authors suggest a potential regulatory mechanism involving specific isoforms for different roles, it is important to establish. Since the authors have the GST-

AP2 constructs, it would also be good to see the reverse pulldown of fig 2b with AP2 as bait, looking for LC3b.

Response: We thank the referee for raising this point, which has led us to reinvestigate whether both appendage domains of AP-2 α are capable of binding to LC3. In the **new Figure 2a** we now show that both appendage domains of AP-2 α can associate with LC3b, although we note a preference of LC3b for AP-2 α_A over AP-2 α_C . Similar results are seen if GST-AP-2 α appendages are used as a bait for affinity purification of native LC3b from mouse brain extracts as shown in the **new Suppl. Fig. S2b**.

Moreover, we have added Coomassie Blue stained gels (in the **new Suppl. Fig. S2a** and **S2c**) to reveal the purity of the recombinant proteins used for these experiments. For the **new Suppl. Fig. S2b** we have also included the Ponceau S stain of the membranes used for immunoblotting and have added the corresponding Ponceau S stain for the experiment shown in Fig. 2e as a **new Suppl. Fig. S2d**.

Further to that, does the AP2 co-IP any of the subunits for anterograde trafficking – potentially would expect not based on the preference of retrograde trafficking but could strengthen that argument.

Response: As shown in the **revised Fig. 2c** using samples from the exact same experiment we are unable to detect the anterograde motor kinesin KIF5A in AP-2 immunoprecipitates from brain. These data argue that AP-2 specifically associates with p150^{Glued}/ dynactin-based retrograde motors, at least under these conditions.

2) Since the authors postulate that this new role for AP2 may be specific to the AP2 α_A isoform, does overexpression of the C isoform fail to rescue the retrograde velocity or the branching phenotype for example? Does it also have the LIR binding site shown to be important in the A isoform?

Response: The question is related to the binding of LC3 to AP-2 α . First, we show in the **new Figure 2a** and **new Supplementary Fig. S2b** that both appendage domains of AP-2 α can associate with LC3b with a preference of LC3b for AP-2 α_A over AP-2 α_C . In all functional studies including our mouse knockout (KO) we have manipulated the endogenous expression of AP-2 μ , not α , because of the possible redundancy of the α_A and α_C isogenes. In fact, based on the presence of the LIR motif in both α_A - and α_C -adaptins (see Tian et al, PNAS 2013) we would expect at least a partial functional redundancy between α_A and α_C .

3) Point for discussion perhaps is the loss of neuronal complexity shown by AP2 KO. The authors show a similar phenotype with ATG5 KO which is involved in the autophagosome maturation pathway but this does not mean that loss of AP2 causes the same phenotype via this pathway. How does this compare to KO of other endocytosis proteins for eg., can the lack of arborisation also be due in part to AP2's role in endocytosis?

Response: The observed phenotype of autophagosome accumulation in axons has not been reported for any other endocytosis mutant, at least to our knowledge. None of this is seen in Stonin 2, AP180 or CALM KO mice studied by ourselves or in dynamin 1 or dynamin 1/2 DKO mice investigated in the De Camilli lab. Moreover, we observe that a specific point mutant of AP-2 α_A within the LIR motif incapable of binding to LC3 but perfectly able to associate with other endocytic proteins supports clathrin-mediated endocytosis but impairs autophagosome transport (please see **Figure 4**). Interestingly, recent data published in *Neuron* (Soukop et al., 2016) and *Cell Reports* (Murdoch et al., 2016) in *Drosophila* and mice have uncovered an endocytosis-independent role for the endocytic protein endophilin in neuronal autophagosome formation, e.g. upstream of the function of AP-2 in autophagosome transport described in this study. These works are now cited and discussed on p. 16 of our revised Ms.

4) Given the well known role of autophagosomes in neurodegeneration, I can understand the authors argument that AP2 KO morphological brain defects seen are attributed to neurodegeneration but I do not feel that the authors have completely excluded the possibility that the defects were caused by a defect in development rather than specifically neurodegeneration. The mouse is missing AP2 which does have essential roles in endocytosis and so arguably could also be responsible for the defects. Were the structures ever there in the first place given that the mice are described as lagging behind in development from birth? I think that this distinction is important since if the structures were not correctly formed, it is not the same as the structures degenerating. Do the authors have any further evidence to support their argument? It does not makes the trafficking conclusions of the paper any less valid or interesting but it may require the neurodegeneration conclusions to be a little less definitive.

Response: We thank the reviewer (see also our response to reviewers 2 and 3) for highlighting this important point that we have tackled in several ways. First, we have carried out additional experiments regarding the question of whether AP-2 loss causes neurodegeneration. First, as kindly suggested by the *editor* we have carried out further histological analysis of the temporal progression of neurodegeneration in conditional AP-2 μ KO. We show in the **new Supplementary Fig. 7g-p** that, although the brain morphology of AP-2 μ -deficient mice appeared normal at p4, already at p7 first signs of neuronal loss were detectable in thalamic nuclei. This was followed by the dramatic appearance of spongiform neurodegeneration in the cortex at p14. Second, we show by Fluoro-Jade staining, a probe that detects dying neurons, that the neurodegeneration in brains of conditional AP-2 μ KO mice at p20 is due to neuronal death (**new Supplementary Fig.7d**) and, further, that this loss of neurons in absence of AP-2 is mediated by apoptosis as evidenced by elevated levels of active caspase-3 in lysates from KO neurons (**new Supplementary Fig. 7e,f**). Taken together these data confirm our initial proposal that AP-2 μ is required to prevent neuronal loss and neurodegeneration.

Finally, related to the above we now show in the **new Supplementary Fig. 6j-l** that conditional loss of AP-2 μ induced at much later time points (e.g. DIV8 instead of DIV0) also impairs neuronal branching complexity of mature neurons, suggesting that loss of AP-2-mediated retrograde autophagosome transport induces post-developmental neurite pruning.

Minor points:

1) *Figure 1: the authors measure the colocalisation of the eGFP-LC3 and mRFP-AP2 with the Pearson's coefficient, but what is the Rp of the empty control vector i.e. mRFP for comparison? Likewise, a +ve interactor for LC3 may also be helpful to get a feel for the extent of colocalization (colocalization of ATG5 GFP+ LC3b RFP).*

Response: We thank the reviewer for this suggestion that we have gladly followed. In the **new Suppl. Fig. S1e,f** we show that the degree of colocalization of LC3b with AP-2 is comparable to that seen for LC3b with the bona fide autophagy component ATG12. Moreover, pixel shifting our images reveals that the colocalization of AP-2 with LC3b clearly is non-random. These additional data are shown in the **new Suppl. Figs. S1c and S1k** and described in the corresponding **legend**.

2) *The authors also show colocalisation with an autophagosome marker, but is there any colocalisation with other things like motor proteins since biochemically there is evidence of binding to motor proteins?*

Response: We refer the referee kindly to **Suppl. Fig. S2**, in which we show colocalization of LC3b with both AP-2 and with p150^{Glued}-dynactin.

Related to that, is there colocalisation with synaptic markers???? I am not disputing that the puncta traffic, but I was interested in the stationary ones, are they at synapses?

Response: In the **new Suppl. Fig. S1g-i** we show that stationary AP-2 puncta are indeed largely confined to presynaptic sites demarcated by the active zone marker Munc-13-1-eYFP.

3) *Several of the quantification graphs where conditions have been normalised prior to plotting, are described in the legends as normalised to 100 % but graph shows normalised to 1. I appreciate it is essentially the same thing but would be easier to understand if they were consistent.*

Response: We have made a strong effort to present non-normalized data whenever possible.

4) *Characterisation of endocytic proteins in the KO cultures is only displayed for AP2 α levels. The remainder of the graphs is of the p21 mouse lysates and this distinction is not clear in the text. The two may be different due to the fact that the mouse developed to p21 without AP2 but the cultures are made from mice which develop normally with AP2 and then the cultures are treated to delete the AP2. Culture lysates could be blotted for the other endocytic proteins for best comparison or at least, clear distinction made in the text.*

Response: We have made the distinction clear in the text. As we do not detect significant changes in the levels of any endocytic proteins studied in brains from postnatal KO mice, in which AP-2 has been deleted during development we feel that it is highly unlikely to see changes in neuronal cultures, in which the KO has been induced acutely by tamoxifen addition.

5) *Could the authors also comment on the anterograde velocity in the KO cells expressing mRFP-eGFP-LC3 since the text describes bidirectional movement? The title of this section focuses on conclusions about retrograde transport but the authors have not excluded an effect on anterograde trafficking.*

Response: We have quantitatively analyzed anterograde transport of LC3b puncta and did not detect significant changes (**new Suppl. Fig.3g**), although there is a tendency towards reduced anterograde velocity, possibly as an indirect consequence of impaired retrograde movement as discussed in the Ms text. We also show that mitochondrial motility is unaltered in absence of AP-2 (**new Suppl. Fig. S3h-k**).

6) *The authors conclude that retrograde transport is reduced in KO neurons by illustrating that the velocity is reduced. Are there the same number of autophagosome puncta moving? I.e. are the proportions of stationary vs retrograde and anterograde similar?*

Response: We show in the **new Suppl. Fig.3e,f** that the number of retrogradely mobile autophagosomes is reduced in AP-2 KO neurons and that, accordingly, the proportion of mobile vs. stationary LC3b puncta is changed.

7) *Folimycin application experiments showing accumulation of LC3b puncta. How does application of folimycin on wt cells compare to AP2 KO cells? Data is shown for these conditions but not together so hard to compare to what extent KO accumulates LC3b puncta in the context of treatments known to cause accumulation of autophagosomes. A side by side comparison would be helpful or the puncta/ μm^2 values instead of/in addition to the normalised data in panel m.*

Response: We have replotted the data as puncta/ μm^2 as suggested in **Fig. 3m**.

8) *The text states that mTORC1 signalling was “slightly upregulated” (results AP2 regulates autophagosome transport and turnover independent of its role in endocytosis line 11). Since the quantification does not show any statistical upregulation, this statement should be toned down a little to reflect a possible trend since the only component that is upregulated is the inactive form of the S6 kinase.*

Response: Thank you. We have rephrased the corresponding description and statement in the Ms text.

Related, when describing the rescue by mutant AP2 in fig 4, the statistics displayed do not show that the mutant is better than the wild type.

Response: We assume the referee might be referring to the transferrin uptake experiment shown in Fig. 4g,h. We have phrased our statement more cautiously that mutant AP-2 α_A is fully capable

of rescuing defective transferrin endocytosis in absence of endogenous AP-2 α . This is overtly the case and statistically significant.

9) *The authors look at accumulation of pTrkB in neurites, what happens to non-phosphorylated TrkB or another receptor such as p75 which is activated by BDNF but does not couple to this pathway? Is this accumulation specific to the activated form or a consequence of a more generalised stalling of trafficking and recycling?*

Response: We have analyzed the levels of p75NGF receptor and did not detect any significant changes as shown in the **new Suppl. Fig. S5k-m**.

10) *Does treatment with BDNF do anything to the retrograde velocity in KO neurons (fig 5)? Arguably probably not since there is no AP2 in the KO but given the experiments that come later with rescued branching, it is important to know.*

Response: We thank the referee for this important point. We have analyzed the effect of BDNF on retrograde transport but as predicted do not detect any significant effect as shown in the **new Fig. 5a,b**.

Reviewer #2:

The manuscript “Retrograde transport of TrkB-containing autophagosomes via the endocytic adaptor AP-2 mediates neuronal complexity and prevents neurodegeneration” by Kononenko et al describes a novel role for the well-characterised adaptor protein AP-2 in dynein-dependent transport of autophagosomes. This is a rather unexpected finding that is well supported by the large amounts of data presented.

Response: We thank the referee for these very positive remarks and for highlighting the novelty of our findings and its support by a large amount of data.

I have some queries/ comments for this section:

Can the authors comment on clathrin recruitment to these autophagosomes?

Response: We have conducted additional experiments to address this interesting point. In the **new Suppl. Fig. S1l,m** we show that endogenous clathrin, indeed fails to be enriched on AP-2-containing autophagosomes, further suggesting that the role of AP-2 in autophagosome transport is independent of its established endocytic function.

What percentage of autophagosomes is AP-2 positive,

Response: Quantitative analysis depicted in the **new Suppl. Fig. S1d** shows that about 70% of all autophagosomes are AP-2-positive.

...and would this suggest a PIP2 enrichment in the autophagosome membrane of these autophagosomes?

Response: We demonstrate in **Figs. 2 and S2** that AP-2 via its α -subunit directly associates with LC3, which likely mediates its recruitment to autophagosomes. As autophagosomes are known to require PI(3)P we do not envision a role for PI(4,5)P₂ in AP-2 recruitment, in agreement also with the lack of clathrin on AP-2-positive autophagosomes (**new Suppl. Fig. S11,m**). However, that said, further studies in the future may need to address this issue in more detail.

Conversely, seeing that the authors observe bidirectional movement of AP-2 positive carriers, but preferentially retrograde movement of autophagosomes, could they speculate on the nature of anterograde AP-2 positive carriers?

Response: We have quantitatively analyzed anterograde transport of LC3b puncta and did not detect significant changes (**new Suppl. Fig.3g**), although there is a tendency towards reduced anterograde transport velocity, possibly as an indirect consequence of impaired retrograde movement as discussed in the Ms text. We also show that mitochondrial motility is unaltered in absence of AP-2 (**new Suppl. Fig. S3h-k**).

Given these data one possibility is that anterograde carriers may deliver AP-2 to distal axons to support its function in endocytosis and retrograde autophagosome transport.

And is there any difference in the total number of autophagosomes in absence vs presence of AP-2?

Response: We show in **Figure 3d-k** that AP-2 KO neurons accumulate LC3-positive autophagosomes. This accumulation is not due to increased formation but most likely reflects impaired retrograde transport as illustrated in **Figure 3a-c** and **Suppl. Fig. S3e-g**.

A minor comment: Could you please move the explanation of folimycin function from p7 to p6 (where it is first mentioned)?

Response: Done - described now on p.5 of the revised Ms.

The authors then go on to discuss that autophagosomes traffic TrkB, and this process is required for BDNF-TrkB signalling. They further state that disruption of this process leads to neurodegeneration. This section would benefit from more detailed descriptions and/or experimental timelines. The authors state that BDNF treatment leads to an increase in the speed RFP-TrkB movement (is this an increase in speed or in % moving vesicles?).

Response: In **Figure 5c** we show that BDNF increases the frequencies of long-distance travels of LC3-TrkB-containing autophagosomes. This corresponds to a significant increase in the mean

retrograde velocity as displayed in **Figure 5d**. Importantly, we now also demonstrate in the new **Suppl. Fig. S5b** that the effect of BDNF stimulation on retrograde transport velocity is lost in AP-2 KO neurons.

What time after treatment is this seen?

Response: The onset of these effects is within 5 min post-BDNF application as indicated also in **Figure 5c**.

Also, overexpression of RTKs frequently leads to auto-activation – could the authors comment please?

Response: We have used overexpression of TrkB-mFRP only for monitoring its axonal transport. While we agree that receptor overexpression may lead to autoactivation none of our functional data relies on TrkB-overexpressing neurons.

Also, the kymographs suggest that AP-2 ablation stops TrkB movement in both directions? Is other transport (eg mitochondria) intact?

Response: We have gladly followed the referee's suggestion to monitor axonal transport of mitochondria. We show in the **new Suppl. Fig. S3h-k** that axonal transport of mitochondria proceeds unperturbed in the absence of AP-2.

The authors go on to show reduced dendritic complexity (have they looked at axons?) in the AP-2 KO, which can be rescued by re-expression.

Response: In our analysis of cultured neurons we have used eGFP transfection to label and quantify both axons and dendrites as described in materials and methods.

It appears to me that the rescue leads to higher complexity than wildtype? Therefore AP-2 overexpression drives dendritic complexity? Please comment.

Response: Our quantitative analysis shown in **Figure 5m** indicates a full rescue of neuronal branching by re-introduction of AP-2 μ . Our subjective impression is that overexpression of AP-2 α_A (WT) may indeed increase branching complexity as shown in **Suppl. Fig. S6a-e**. However, as we did not rigorously test this side-by-side we prefer not make any definitive statements regarding this point in our Ms.

Also, please give experimental details/timelines on when cre was activated and when the rescue constructs were transfected.

Response: We apologize for this omission: We now quote this information in the results and provide details in the materials and methods section and in the legends. For rescue experiments, the AP-2(μ) KO was induced by tamoxifen addition at DIV0, rescue plasmids were introduced at DIV8, and neurons were analyzed at DIV14.

Is this a developmental effect or pruning?

Response: We thank the reviewer raising this important point. We now show in the **new Supplementary Fig. 6j-l** that conditional loss of AP-2 μ induced at much later time points (e.g. DIV8 instead of DIV0) impairs neuronal branching complexity of mature neurons, suggesting that loss of AP-2-mediated retrograde autophagosome transport induces post-developmental neurite pruning.

What DIV are the cultures?

Response: We apologize for this omission: For AP-2 KO neurons DIV14-15, for ATG5 DIV20-21 as described now in the text and in the materials and methods section.

Similar for the in vivo data: I presume this is a snapshot of one time point (seeing how rapidly the KOs succumb). At what age are dendritic complexity and gross brain morphology assessed?

Response: Most data are from p20 animals. In response to the referees we have also analyzed additional time points (p3, p7, p14) that are now clearly described in the text and in the materials and methods section.

The claim that this is “degeneration” (eg manuscript title; p11) would need further time points to show reduction over time – seeing the rather young age at which the mice die this might be a developmental defect? Please clarify or add time points.

Response: We thank the reviewer (see also our response to reviewers 1 and 3) for highlighting this important point that we have tackled in several ways. First, we have carried out additional experiments regarding the question of whether AP-2 loss causes neurodegeneration. First, as kindly suggested by the *editor* we have carried out further histological analysis of the temporal progression of neurodegeneration in conditional AP-2 μ KO. We show in the **new Supplementary Fig. 7g-o** that, although the brain morphology of AP-2 μ -deficient mice appeared normal at p4, already at p7 first signs of neuronal loss were detectable in thalamic nuclei. This was followed by the dramatic appearance of spongiform neurodegeneration in the cortex at p14. Second, we show by Fluoro-Jade staining, a probe that detects dying neurons, that the neurodegeneration in brains of conditional AP-2 μ KO mice at p20 is due to neuronal death (**new Supplementary Fig.7d**) and, further, that this loss of neurons in absence of AP-2 is mediated by apoptosis as evidenced by elevated levels of active caspase-3 in lysates from KO neurons (**new Supplementary Fig. 7e,f**). Taken together these data confirm our initial proposal that AP-2 μ is required to prevent neuronal loss and neurodegeneration.

Finally, related to the above we now show in the **new Supplementary Fig. 6j-l** that conditional loss of AP-2 μ induced at much later time points (e.g. DIV8 instead of DIV0) also impairs neuronal branching complexity of mature neurons, suggesting that loss of AP-2-mediated retrograde autophagosome transport induces post-developmental neurite pruning.

Finally, the authors show reduced levels of BDNF in the AP-2 KO mice, and conclude that the morphological defects are due to lack of positive feedback. I am a bit lost here. Have the authors looked at the induction of other IEGs downstream of BDNF/TrkB (eg Arc)?

Response: We thank the referee for this suggestion. In the **new Figure 7d** we demonstrate that loss of AP-2 indeed also causes a reduction in the mRNA levels of BDNF, in agreement with recent findings (Cheng et al., PNAS 2011; Tuvikene et al., J. Neurosci 2016).

How does replenishing BDNF lead to rescue – what is downstream? And how is retrograde transport required in mass culture (fig 5), but not if exogenous BDNF is added?

Response: We reasoned that defective arborization of AP-2 μ KO neurons should be rescued by boosting BDNF signalling through exogenous bath application of BDNF, rather than through local secretion and activation of axonal TrkB receptors. In this setting bath application of BDNF in mass cultures of AP-2 KO neurons is expected to activate soma-confined TrkB receptors, thereby eliminating the necessity for retrograde transport along the axon. This is exactly what we observed and show in **Figure 7e,f**.

Also, it is rather curious that the authors do not find an otherwise well-documented effect of BDNF on neuronal complexity in wildtype neurons (eg work of McAllister, Cohen-Cory etc). Please explain.

Response: What is documented in these works is mostly the increase in the number and the length of proximal dendrites, a process that we have not analyzed in any detail here. Several works have shown that BDNF does not have an influence on the number of branching points (Dijkhuizen & Ghosh, 2004; Kwon et al., 2011). We now cite both of these works in the revised Ms text.

It would be nice to see the full BDNF blot (or ELISA) in figure 7b to assess whether it is BDNF levels or processing that is defective.

Response: Thank you. We now show the full blots illustrating the levels of pro-BDNF and mature BDNF in the **new Suppl. Fig. 7v** and quantify these in the **new Suppl. Fig. 7w**.

Also, how does a receptor signal once within the autophagosomal double membrane structure?

Response: This is an interesting point that deserves further investigation in the future. We think that TrkB is unlikely to signal from within autophagosomes, but may release activated kinases/ and other effectors from autophagosomes as these undergo fusion with lysosomes in the cell body, resulting in "delivery" of the active kinase or effector to the soma, while active TrkB may be degraded. This hypothetical scenario will require extensive testing in future studies.

Reviewer #3:

In this manuscript, Kononenko et al. investigated a novel mechanism of transport of TrkB-containing autophagosomes via the endocytic adaptor protein AP-2. They used biochemical, cell based and in vivo studies to provide largely strong evidence for this mechanism.

Response: We thank the referee for these very positive remarks and for highlighting the strength of our experimental data.

What is less compelling is the relevance to neurodegeneration as the deficits they observe could also be developmental.

Response: We thank the reviewer (see also our response to reviewers 1 and 2) for highlighting this important point that we have tackled in several ways as detailed further below.

Specific comments:

Fig. 3, Panel d: EM shows enlarged dense vesicular bodies with concentric multilamellar structures. These could be lysosomes or late stage amphisomes. The cigar-shaped structure in the KO image which looks like a mitochondrion also seems enlarged and dense. Is the magnification the same?

Response: These images are indeed the same magnification. We did not note any enlargement of mitochondria.

Further to this point we have analyzed mitochondrial transport in axons and show in the **new Suppl. Fig. S3h-k** that axonal transport of mitochondria proceeds unperturbed in the absence of AP-2.

Please provide additional images.

Response: We provide additional images in the **revised Fig. S3o** now.

How do the lysosomes in the cell soma (around the nucleus) look like in wild-type and AP2 KO neurons?

Response: As suggested by the reviewer we have analyzed the number of lysosomes by light and electron microscopy. In the **new Suppl. Fig. S3l-n** we provide example images and

quantifications to show that the number of lysosomes is unchanged in AP-2 KO neurons, in agreement with analysis by light microscopy displayed in **Suppl. Fig. S4c,d**.

Fig. 3, Panel m: treatment with folimycin caused equal accumulation of LC3b-positive puncta in AP2 KO neurons compared to controls. However, if the transport to the lysosome is delayed in AP2 KO neurons, one might expect to see less accumulation of LC3b/Rab7 puncta over the duration of the experiment. Please clarify.

Response: The primary purpose of the folimycin experiments was to determine whether the observed accumulation of LC3- and Rab7-positive autophagosomes may be due to elevated autophagosome formation rather than delayed transport and fusion with lysosomes in the soma. As autophagosomes form in the periphery (e.g. in distal axons) the observed phenotype in our view reflects decreased degradation of LC3b/Rab7 positive autophagosomes due to defective retrograde autophagosome transport rather than increased synthesis. We do not expect to see less accumulation of LC3b/ Rab7 puncta over the time course of the experiment (4 hours in this case) as new autophagosomes likely are being formed.

In this regard, why did the authors measure starvation-induced flux in Fig 5, panel b and panel f rather than flux with and without a lysosomotropic inhibitor?

Response: We kindly refer the referee to the data shown in **Figure 3m,n**, where we have analyzed autophagic flux in the presence or absence of the lysosomotropic inhibitor folimycin.

How long was the starvation? I was unable to find this information.

Response: The starvation was for 3 hours as detailed in the revised materials and methods section. We apologize for the omission.

Fig. 3, Panel n is missing from the figure

Response: We apologize for this error that has been corrected in the revised version of our Ms.

Please include Rab5 (early endosome marker) as an additional control for immunofluorescence to show whether early endosomes change in AP2 KO mice?

Response: In the **new Suppl. Fig. S4a,b** we have analyzed Rab5 positive early endosome levels but did not observe any significant changes in AP-2 KO neurons.

Fig 6: The data presented here are interesting but do not distinguish between a developmental and truly degenerative phenotype. Therefore caution is required when implicating this pathway in neurodegeneration.

Response: As noted above we have tackled this important point in several ways. First, we have carried out additional experiments regarding the question of whether AP-2 loss causes neurodegeneration. First, as kindly suggested by the *editor* we have carried out further histological analysis of the temporal progression of neurodegeneration in conditional AP-2 μ KO. We show in the **new Supplementary Fig. 7g-o** that, although the brain morphology of AP-2 μ -deficient mice appeared normal at p4, already at p7 first signs of neuronal loss were detectable in thalamic nuclei. This was followed by the dramatic appearance of spongiform neurodegeneration in the cortex at p14. Second, we show by Fluoro-Jade staining, a probe that detects dying neurons, that the neurodegeneration in brains of conditional AP-2 μ KO mice at p20 is due to neuronal death (**new Supplementary Fig.7d**) and, further, that this loss of neurons in absence of AP-2 is mediated by apoptosis as evidenced by elevated levels of active caspase-3 in lysates from KO neurons (**new Supplementary Fig. 7e,f**). Taken together these data confirm our initial proposal that AP-2 μ is required to prevent neuronal loss and neurodegeneration.

Finally, related to the above we now show in the **new Supplementary Fig. 6j-l** that conditional loss of AP-2 μ induced at much later time points (e.g. DIV8 instead of DIV0) also impairs neuronal branching complexity of mature neurons, suggesting that loss of AP-2-mediated retrograde autophagosome transport induces post-developmental neurite pruning.

The accumulation of p62 puncta both in vivo and in cells suggest a broader defect in autophagic degradation. How do the authors explain this if AP2 is more selectively involved in the pathway they describe?

Response: While our data support a general role of AP-2 retrograde transport of autophagosomes containing TrkB as one of their cargos (see **Figure 5** and **Suppl. Fig. S5**), we consider it likely that other autophagic cargos may also be affected by loss of AP-2 as illustrated in part by the accumulation of p62. Future studies will be needed to define these additional cargos in more detail.

BDNF addition accelerates the transport of LC3/TrkB carriers. However, the data linking LC3/TrkB trafficking to BDNF transport and its nuclear signalling is weak and could be strengthened by studies using (1) labeled BDNF in WT and AP2 KO neurons and (2) addition of BDNF in WT and AP KO neurons cultured in microfluidic devices where one would expect that the KO neurons would not respond to BDNF added at the synaptic terminal compartment.

Response: Unfortunately, we have failed to achieve a uniform AP-2 KO in a microfluidic system. This is due to the fact that once plated, the neurons in the somato-dendritic compartment are not easily accessible. However, our neuronal cell culture protocol includes two rounds of media exchange after plating: On DIV1, where half of the media are removed and tamoxifen-containing media are added and on DIV2, when 1 ml of tamoxifen-free medium is added. As equal solution interchange is impossible in microfluidic devices, the use of microfluidic chambers may be difficult to combine with tamoxifen treatment of neuronal mass cultures used in our study. Labelled BDNF is the perfect tool to be used in microfluidic chambers, however, in our mass cultures uptake of BDNF-activated TrkB receptors will take place all over the cell, including the soma and dendrites,

and thus, will not help to answer the question of retrograde TrkB trafficking in axons of AP-2 KO neurons.

To further support our notion that retrograde transport of TrkB is required for signalling, we now demonstrate in the **new Figure 7d** that loss of AP-2 causes a reduction in the mRNA levels of BDNF, an established transcriptional target of TrkB signalling (Cheng et al., PNAS 2011; Tuvikene et al., J. Neurosci 2016).

REVIEWERS' COMMENTS:

Reviewer #1 (Remarks to the Author):

I would like to thank the authors for including all the new data and for modifying some of the data previously included. I think that the main question of whether or not neurodegeneration was a valid conclusion is definitely strengthened with the new data. The authors have answered my other concerns satisfactorily too.

Reviewer #2 (Remarks to the Author):

The authors of the manuscript "Retrograde transport of TrkB containing autophagosomes via the endocytic adaptor AP-2 mediates neuronal complexity and prevents neurodegeneration" have thoroughly addressed all my comments and I have no further queries and no concerns. I am sure the study will be of great interest to the neuroscience and wider cell biology communities.

Reviewer #3 (Remarks to the Author):

The authors have addressed all my concerns

Response to the reviewers

(reviewers' comments are given in italics, our response to each point is given below).

We would like to thank all three referees for their support regarding publication of our manuscript in *Nature Communications*. No further actions were necessary.

Reviewer #1:

I would like to thank the authors for including all the new data and for modifying some of the data previously included. I think that the main question of whether or not neurodegeneration was a valid conclusion is definitely strengthened with the new data. The authors have answered my other concerns satisfactorily too.

Response: Thank you.

Reviewer #2:

The authors of the manuscript "Retrograde transport of TrkB containing autophagosomes via the endocytic adaptor AP-2 mediates neuronal complexity and prevents neurodegeneration" have thoroughly addressed all my comments and I have no further queries and no concerns. I am sure the study will be of great interest to the neuroscience and wider cell biology communities.

Response: We thank the reviewer for his support and these encouraging statements.

Reviewer #3:

The authors have addressed all my concern.

Response: Thank you.